# Chemotherapy Resistance: Role of Mitochondrial and Autophagic Components

**DOI:** 10.3390/cancers14061462

**Published:** 2022-03-12

**Authors:** Entaz Bahar, Sun-Young Han, Ji-Ye Kim, Hyonok Yoon

**Affiliations:** 1College of Pharmacy and Research Institute of Pharmaceutical Sciences, Gyeongsang National University, Jinju 52828, Korea; entazbahar@gnu.ac.kr (E.B.); syhan@gnu.ac.kr (S.-Y.H.); 2Department of Convergence Medical Science and Biochemistry, Institute of Health Sciences, College of Medicine, Gyeongsang National University, Jinju 52727, Korea; 3Department of Pathology, Ilsan Paik Hospital, College of Medicine, Inje University, Goyang 10380, Korea; alucion@paik.ac.kr

**Keywords:** chemotherapy resistance, mitochondria, mitophagy

## Abstract

**Simple Summary:**

Chemotherapy resistance is a common occurrence during cancer treatment that cancer researchers are attempting to understand and overcome. Mitochondria are a crucial intracellular signaling core that are becoming important determinants of numerous aspects of cancer genesis and progression, such as metabolic reprogramming, metastatic capability, and chemotherapeutic resistance. Mitophagy, or selective autophagy of mitochondria, can influence both the efficacy of tumor chemotherapy and the degree of drug resistance. Regardless of the fact that mitochondria are well-known for coordinating ATP synthesis from cellular respiration in cellular bioenergetics, little is known its mitophagy regulation in chemoresistance. Recent advancements in mitochondrial research, mitophagy regulatory mechanisms, and their implications for our understanding of chemotherapy resistance are discussed in this review.

**Abstract:**

Cancer chemotherapy resistance is one of the most critical obstacles in cancer therapy. One of the well-known mechanisms of chemotherapy resistance is the change in the mitochondrial death pathways which occur when cells are under stressful situations, such as chemotherapy. Mitophagy, or mitochondrial selective autophagy, is critical for cell quality control because it can efficiently break down, remove, and recycle defective or damaged mitochondria. As cancer cells use mitophagy to rapidly sweep away damaged mitochondria in order to mediate their own drug resistance, it influences the efficacy of tumor chemotherapy as well as the degree of drug resistance. Yet despite the importance of mitochondria and mitophagy in chemotherapy resistance, little is known about the precise mechanisms involved. As a consequence, identifying potential therapeutic targets by analyzing the signal pathways that govern mitophagy has become a vital research goal. In this paper, we review recent advances in mitochondrial research, mitophagy control mechanisms, and their implications for our understanding of chemotherapy resistance.

## 1. Introduction

Chemotherapy resistance is a prevalent phenomenon during cancer treatment that cancer researchers are working to understand and conquer [1]. It alters cancer cell signals that are needed for cellular growth or immune response stimulation through a wide range of molecular pathways [2,3]. In our previous literature review and experimental study, we revealed that endoplasmic reticulum stress-dependent signaling contributes to chemotherapy resistance [4,5,6]. Mitochondria are a key intracellular signaling hub that are emerging as important determinants of several aspects of cancer’s development and progression, including metabolic reprogramming, acquisition of metastatic capability, and response to chemotherapeutic drugs. Mitochondrial dysfunction has been linked to the onset and progression of cancer. However, recent evidence suggests that pathogenic mutations or mitochondrial genome depletion may play a role in the development of chemoresistance in cancerous tumors. Despite the fact that mitochondria are well-known for their role in cellular bioenergetics, coordinating adenosine triphosphate (ATP) production from cellular respiration, little is known about their involvement and activity in chemoresistance [7].

Mitophagy, or selective autophagy of mitochondria, is important for quality control because it can efficiently degrade, remove, and recycle defective or damaged mitochondria [8]. In many cancers, severe mitophagy abnormalities are linked to total impairment of mitochondrial functioning, oncogenesis, and tumor development. In addition, mitophagy serves a dual role in cancer therapy, inducing cell death while also promoting cell survival [9]. According to a recent study, mitophagy influences both the efficacy of tumor chemotherapy and the degree of drug resistance, because cancer cells use mitophagy to rapidly sweep away damaged mitochondria in order to mediate their own drug resistance [10]. Although substantial progress has been made in defining mitophagy regulation in mitochondria and the role of mitophagy in chemoresistance, understanding how mitophagy acts as a chemo-resistant in cancer therapy remains an unsolved topic. Therefore, in this review, the role of autophagy and mitochondrial functionality in determining chemotherapeutic resistance will be discussed in light of recent research.

## 2. Potential Conventional Mitochondrial Targets for Chemotherapy

Mitochondria play a critical role in the generation of metabolic energy in eukaryotic cells, and they are responsible for the energy derived from the breakdown of carbohydrates and fatty acids, which is converted to ATP by the process of oxidative phosphorylation (OXPHOS) [11,12]. Most mitochondrial proteins are translated on free cytosolic ribosomes and imported into the organelle by specific targeting signals [13]. In addition, mitochondria are unique among the cytoplasmic organelles already discussed in that they contain their own deoxyribonucleic acid (DNA) called mtDNA, which encodes transfer ribonucleic acids (tRNAs), ribosomal ribonucleic acids (rRNAs), and some mitochondrial proteins [14]. The assembly of mitochondria thus involves proteins encoded by their own genomes and translated within the organelle, as well as proteins encoded by the nuclear genome and imported from the cytosol [15]. Since mitochondria have distinct structural and functional properties, therapies designed to influence the activity of this organelle can be targeted specifically for therapeutic benefit. The mitochondrion is engaged in a number of vital functions in eukaryotic cells, including energy metabolism and cell cycle regulation, which is critical in cancer physiology. The argument for developing anticancer therapies based on mitochondrial targeting is backed up by substantial evidence. Molecules found on or inside mitochondria are regarded as excellent pharmacological targets, and many attempts are being made to exploit these targets in order to develop tailored therapeutics for a variety of disorders, including cancer [16]. For example, mitochondria are recognized to play a significant part in the complicated apoptotic pathway, causing cell death by a variety of ways such as interrupting electron transport and energy metabolism, releasing or activating apoptotic proteins, and changing the cellular redox potential [17,18,19,20]. Therefore, mitochondrial drug-targeting approaches could create more opportunities for manipulating metabolic activity and allowing for the selective protection or destruction of cells for therapeutic benefit in a range of disorders (Figure 1).

### 2.1. Mitochondrial Calcium Ion (Ca^2+^)

Calcium ion (Ca^2+^) signaling is necessary for a range of cell processes; hence, it must be maintained in check to ensure proper cellular homeostasis [21,22]. Mitochondria are important organelles in the control of Ca^2+^ homeostasis in cells. Although Ca^2+^ regulates mitochondrial respiration, which is dependent on ATP synthesis, excess Ca^2+^ can cause mitochondrion-mediated death [23]. Mitochondrial Ca^2+^ signaling pathways are receiving more attention due to their diverse role in regulating cell survival and death, and the proteins implicated in those processes may represent alternate targets for cancer therapy [24,25]. The outer mitochondrial membrane (OMM), which is engaged in the permeabilization process, and the inner mitochondrial membrane (IMM) channel are in charge of maintaining and adapting membrane potential (ΔΨ); hence, controlling mitochondrial respiration and reactive oxygen species (ROS) production efficiency [26,27]. As a result, the ion channels residing both in the mitochondrial membranes may have an impact on death resistance.

#### 2.1.1. The Voltage-Dependent Anion Channel (VDAC)

The voltage-dependent anion channel 1 (VDAC1), the most abundant protein on the outer membrane of mitochondria, is one of the major proteins that govern mitochondrial function [28,29,30,31]. VDAC1 is the gatekeeper for the passage of metabolites, nucleotides, and ions; it regulates apoptosis through interactions with apoptotic and anti-apoptotic proteins, including the BCL-2 family of proteins and hexokinase [32,33,34,35]. To coordinate cellular metabolism and maintain OMM integrity, Ca^2+^ from the sarcoplasmic reticulum (SR) or endoplasmic reticulum (ER) enters mitochondria via the VDAC, which is positioned inside the mitochondrial-associated membrane (MAM) domain near the SR [36,37]. VDAC1 is thought to selectively convey apoptotic Ca^2+^ signals from the SR to the mitochondria [29,38,39]. In addition, VDACs were assumed to be a component of the mitochondrial permeability transition pore (mPTP), a giant channel complex implicated in cell protection and cell death, together with other mitochondrial proteins (ANT, PiC, and TSPO) [40,41,42,43]. Moreover, VDACs are thought to be important in the release of ROS, providing an anchoring site for HK binding, cytochrome c release caused by apoptosis, and (4) interacting with the BCL-2 family of proteins [44,45,46,47,48]. However, Parkin can be recruited to defective mitochondria by VDACs, which promotes mitochondrial autophagy [49].

#### 2.1.2. The Mitochondrial Calcium Uniporter (MCU) Complex

The activities of the mitochondrial calcium uniporter (MCU) complex, which consists of the pore-forming subunit of the MCU channel as well as many regulatory proteins (MICU1, MICU2, MICU3, MCUR1, MCUb, and EMRE), are commonly regarded as a mechanism for Ca^2+^ entrance into mitochondria [50,51,52]. MCU plays a crucial function in the regulation of aerobic metabolism, ATP production and cell viability by modulating mitochondrial Ca^2+^ concentration [53,54,55]. In HEK293T and HeLa cells, MCUR1 knockdown (identical to MCU knockdown) inhibits mitochondrial Ca^2+^ absorption, impairs OXPHOS, reduces cellular ATP, and triggers AMPK-dependent pro-survival autophagy [56]. MICU1 promotes cell survival by acting as a gatekeeper for MCU-mediated mitochondrial Ca^2+^ uptake [57,58]. MCU or MCUR1 deficiency impairs the MCU heterooligomeric complex, altering mitochondrial bioenergetics, cell proliferation, and migration, as well as initiating autophagy via AMPK [59,60,61].

### 2.2. Mitochondrial Permeability Transition Pore Complex (mPTPC)

The mitochondrial permeability transition pore complex (mPTPC), a multiprotein complex produced at the contact region between the mitochondrial inner and outer membranes, regulates mitochondrial membrane permeabilization, which occupies a central role as a selective target for anti-cancer therapy [62,63]. Mitochondrial permeability transition (mPT) is a key process in the control of cell death and survival, and is a potential target for cytotoxic drug development [64,65,66]. Alteration in mPT by chemotherapy might influence cell destiny by altering the physiological activity of mPTPC. As a result, it is suggested that forcing the mitochondrial permeability transition pore (mPTP) to open or close for an extended period of time is toxic to the cell [26,67]. For example, pretreatment of mPT with a pore inhibitor, Cyclosporin A (CsA), or a pore opener sensitizer, PK11195, has been discovered to increase mitochondrial apoptosis triggered by resveratrol, a natural polyphenol [68]. It has been demonstrated that modulation of mPT affects multidrug resistance in human hepatocellular carcinoma cells [69].

### 2.3. ATP Synthase and Mitochondrial Reactive Oxygen Species (mtROS)

ATP synthase is a mitochondrial energy conversion enzyme that is found in the inner mitochondrial membrane and is required for OXPHOS under normal conditions [70,71]. In mammals, ATP synthase is responsible for the majority of cellular ATP generation, and its biogenesis pathway can be affected by either structural changes or insufficient quantities [72,73]. The ATP synthase inhibitor causes cytotoxicity in breast and lung cancer cells but not in healthy cells, suggesting that it could be employed as a cancer treatment candidate [74,75]. It has been demonstrated that combination therapy targeting ectopic ATP synthase and 26S proteasome induces ER stress in breast cancer cells [76]. Various anticancer drugs that target mitochondria diminish ATP levels and/or increase ROS generation [77,78,79]. Oligomycin, an inhibitor of the reversible mitochondrial ATP synthase, induces cell death which is thought to be due to ATP depletion caused by inhibition of the FOF1-ATP synthase (FO and F1 are the main subunit of ATP synthase) enzyme complex [77,80]. Citreoviridin inhibits the growth and proliferation by targeting the F1-ATP synthase in breast and lung cancer cells [75,76]. Polyphenolic phytochemicals including quercetin, genistein, and resveratrol inhibit mitochondrial proton FOF1-ATPase/ATP synthase, which has health benefits as well as possible cytotoxicity [77].

Mitochondria are primarily responsible for energy production, but they also regulate the levels of mitochondria-derived reactive oxygen species (mtROS) [26,81]. Superoxide anion (O_2_^•−^), hydroxyl radical (^•^OH), hydrogen peroxide (H_2_O_2_), and singlet oxygen (^1^O_2_) are all examples of mtROS [26,82,83,84,85]. In mitochondria, mtROS are largely produced by electron transport chain (ETC) complex I and III; however, increased ROS is also caused by inhibition of complex IV or V (ATP synthase) [86,87]. Inhibition of ATP synthase causes a rise in mitochondrial membrane potential (ΔΨm), resulting in increased electron leak to superoxide [88]. The mtROS are required for cell proliferation and cancer growth, but they can also cause DNA damage, protein oxidation, and lipid peroxidation, which can lead to cell death [88,89,90,91,92]. Drug options to combat mtROS activity as a cancer treatment are still being developed, as it can both aid and inhibit tumor cell proliferation. The antioxidant system is engaged to ensure that ROS levels do not reach a deadly threshold, while tumor cell lines increase their ROS production [93,94,95,96,97,98,99]. As a result, while some studies have found dietary antioxidants to be useful in treating cancer, others have found them to be ineffective or even harmful. For instance, carotenoids may increase mortality in breast cancer patients, while vitamin C and E intake have been linked to a lower recurrence rate in the same patients [100,101]. Vitamin C enhances the anti-proliferative activity of doxorubicin in breast cancer patients, while high vitamin D levels are linked to improved survival in colorectal cancer patients [102,103].

### 2.4. Mitochondrial DNA (mtDNA)

Mammalian mitochondria have their own mtDNA, which encodes 13 oxidative phosphorylation complex polypeptides, 12S and 16S rRNAs, and 22 tRNAs necessary for mitochondrial activity [104]. The mtDNA, a tiny circular chromosome found inside mitochondria, functions as a biomarker of mitochondrial dysfunction and has been linked to the progression of several diseases, including cancer, suggesting that it may be used as a chemotherapeutic target [105]. It has been reported that mtDNA acts as the critical target of cisplatin in its anticancer activity [106,107]. For example, cisplatin binds to mtDNA and VDAC protein in the mitochondrial membrane to induce apoptosis in head and neck squamous cell carcinoma (HNSCC) [108]. Cisplatin induced mitochondrial DNA damage, impaired respiratory activity, increased oxidative stress, and activated caspase-9 in cultured hippocampal neurons and hippocampal neural stem cells (NSCs) [109]. In cisplatin-treated cells, mtDNA damage is visible, and cells with depleted mtDNA derived from a cisplatin-sensitive ovarian cancer cell line developed resistance to cisplatin-induced cell death [110,111].

### 2.5. DNA Polymerase Subunit Gamma (PolG)

The only DNA polymerase known in human mitochondria, PolG, contains a large catalytic component, PolG1 (140 kDa), as well as two smaller identical accessory subunits, PolG2 (55 kDa), both of which are essential for embryonic development [112,113,114]. The PolG1 mutations have been linked to mtDNA depletion, mitochondrial dysfunction, aging, cancer, and a variety of other illnesses [115,116]. The PolG gene mutations have been found in human malignancies, suggesting that PolG may have a role in decreasing OXPHOS and boosting tumorigenicity in tumor cells [115]. The POLG1 has been shown to have altered genetic and epigenetic regulation in human malignancies, implying that POLG1 germline variations may play a role in tumorigenic characteristics [117]. For example, the breast tumorigenesis is aided by mutations in mitochondrial DNA POLG1 [115]. It is also a crucial component in mtDNA replication and repair [118,119]. It has been reported that targeting mitochondrial DNA PolG could be a unique therapy method for acute myeloid leukemia [120,121]. For examples, in acute myeloid leukemia, the thymidine dideoxynucleoside analog alovudine inhibits mitochondrial DNA polymerase, hinders OXPHOS, and promotes monocytic differentiation [120]. 

### 2.6. Mitochondrial Ribosome and Ribosomal Proteins (MRP)

The mitochondrial ribosome is the organelle in eukaryotic cells that synthesize 13 inner membrane proteins of the oligomeric complex required for OXPHOS [122,123]. Mitochondrial ribosome is essential for ribosomal RNA processing, modification, and binding to ribosomal proteins and cellular respiration, a process linked to cell growth and proliferation [124]. Several mitochondrial ribosomal proteins, notably the death-associated proteins 3 (DAP3) and programmed cell death protein 9 (PDCD9), have been identified as apoptosis-inducing agents in several investigations [125,126]. The mitochondrial ribosomal protein L41 (MRPL41) suppresses tumor cell growth through inducing p53-induced mitochondrion-dependent apoptosis, whereas the activation of p38 mitogen-activated protein kinase (MAPK) and c-Jun N-terminal kinase (JNK) signaling by the S29 ribosomal protein (RPS29) causes apoptosis in a human laryngeal carcinoma cell line [127,128]. The X-linked ribosomal protein S4 (RPS4X), which is involved in cellular translation and proliferation in breast cancer cell lines, has been identified as a partner of the overexpressed multifunctional protein YB-1, and RPS4X depletion leads to consistent resistance to cisplatin [129]. It has been revealed that RPS4X deletion in ovarian cancer cell lines decreased their proliferative rate but more critically, increased their cisplatin resistance [130]. Higher expression of mitochondrial ribosomal protein L65 (MRPL65), also known as PDCD9 or mitochondrial ribosomal protein S30 (MRPS30) in murine fibroblasts, can cause apoptosis by activating the JNK1 and upregulating other transcription factors [131,132].

### 2.7. Mitochondrial Proteins

Apoptosis of cancer cells is a desirable result of chemotherapeutic treatment. Inducing apoptosis by targeting mitochondrial proteins is one of the strategies through which chemotherapeutics kill cancer cells. Mitochondrial apoptosis is regulated by the B-cell lymphoma 2 (BCL-2) family proteins, which are classified into three groups based on their primary function, including (i) anti-apoptotic proteins (BCL-2, BCL-xL, BCL-W, MCL-1, etc.), (ii) pro-apoptotic pore-formers (Bax, Bak, BOK, etc.), and (ii) pro-apoptotic BH3-only proteins (Bad, Bid, Bik, Bim, Bmf, HRK, PUMA, etc.) [133,134,135]. The BCL-2 protein family regulates cell death mainly through direct binding interactions that govern mitochondrial outer membrane permeabilization (MOMP), which result in the irreversible release of intermembrane space proteins, caspase activation, and apoptosis [133,136]. One of the aims of chemotherapy is to induce apoptosis by targeting BCL-2 family proteins and creating MOMP. The MOMP causes the release of pro-apoptogenic substances (such as cytochrome c and Smac/DIABLO) from the mitochondrial intermembrane space (IMS) into the cytosol, which activates a caspase cascade that demolishes and kills the cell [136,137,138]. Hundreds of proteins essential for appropriate cellular function and homeostasis are cleaved by caspases, resulting in the biochemical and morphological characteristics of apoptosis [139,140]. Practically all of the developed targeted medicines that have the potential to be lethal to cancer cells only, rather than normal healthy cells, work by activating the same mitochondrial apoptotic pathways as standard chemotherapeutic agents [141,142,143,144].

## 3. Novel Metabolic Switch Targeted Anti-Cancer Therapeutic Approaches

### 3.1. Oxidative Phosphorylation (OXPHOS)

Evidence shows that mitochondrial metabolism is critical in cancer cell survival and growth [145]. Glycolysis and OXPHOS are two of the most important metabolic processes for cells to obtain energy [146]. Even when sufficient oxygen is available, cancer cells prefer to employ glycolysis, an inefficient metabolic pathway for energy metabolism that promotes carcinogenesis and cancer growth [147,148]. Glycolysis is significantly simpler than OXPHOS, often known as aerobic mitochondrial respiration. In normal cells, metabolic activities depend mostly on mitochondrial OXPHOS to create ATP, while cancer cells have increased glycolysis and decreased OXPHOS capability (Figure 2). Aerobic glycolysis is used by cancer cells as a metabolic adaptation relative to normal cells, leading some to believe that OXPHOS is downregulated in most of the cancers [149]. Cancer cells typically create pyruvate through the glycolysis route, which produces lactic acid instead of entering mitochondria and converting to acetyl-CoA to produce ATP in hypoxic conditions [149]. Even in the presence of abundant oxygen, cancer cells preferentially use glycolysis to create ATP, known as aerobic glycolysis or the very well-known Warburg effect [150]. In this process, cancer cells would utilize amino acids, fatty acids, and lipid substances to increase their own proliferation in addition to decomposing glucose for ATP [151]. The unique mitochondrial physiology potentially offers anticancer properties because cancer cells reduce their function by utilizing glycolysis [152]. Therefore, restricting cancer growth by targeting several enzymes involved in aerobic glycolysis has emerged as a potential therapy [153,154]. For example, fasentin is an anticancer agent that works by targeting the glucose transporter (GLUT) and preventing glucose uptake [155]. Treatment of pancreatic CD133+ cancer cells with a selective GLUT1 inhibitor, STF-31, increased the sensitivity to cytotoxic drugs such as gemcitabine, paclitaxel, and 5-fluorouracil, resulting in extensive cell death [156]. In many hexokinase 2 (a key enzyme of aerobic glycolysis) overexpressed malignant tumors, lonidamine, 3-bromopyruvate (3-BrPA) and 2-deoxy-D-glucose (2-DG) have been used as effective hexokinase inhibitors [157,158,159,160,161]. Clinical trials have demonstrated that combining lonidamine with a chemotherapy, such as doxorubicin, improves therapeutic efficacy for breast, prostate, melanoma, brain, and ovarian malignancies [162,163].

Conventionally, it has been suggested that tumor cell proliferation is linked to a wide range of OXPHOS deficits and decreased ATP availability. However, some recent studies have shown that OXPHOS can be also upregulated in certain cancers, even in the face of active glycolysis [164,165]. While OXPHOS produces the most ATP, limiting metabolism to glycolysis provides the biomolecule precursors needed by tumors to maintain a high level of growth [166,167,168,169,170,171,172,173]. OXPHOS is thought to be a process implicated in a variety of malignancies, and it is being investigated as a possible target for cancer treatment. Blocking mitochondrial biogenesis, utilizing drugs that impact mitochondrial function, or using direct high-affinity inhibitors of respiratory chain complexes are all options for inhibiting OXPHOS. The OXPHOS inhibitors are used in various tumors, such as metformin for prostate cancer, phenformin for melanoma, carboxyamidotriazole orotate for glioblastoma, and atovaquone for acute promyelocytic leukemia [174,175]. However, due to inherent or acquired resistance of cancer cells to OXPHOS suppression, there are only a few cases in which OXPHOS inhibition would be sufficient to permanently suppress tumorigenesis. Therefore, the combination therapies could be devised to enhance the efficacy of OXPHOS inhibition in cancer treatment. BRAF inhibition, for example, can be coupled with phenformin to slow the progression of melanoma, and the cKIT inhibitor imatinib can be used with VLX600 to decrease the growth of gastrointestinal stromal tumors in mice [176]. The glucose oxidation checkpoint pyruvate dehydrogenase (PDH) and its inhibitor pyruvate dehydrogenase kinase (PDK) can be targeted to modify glucose oxidation and OXPHOS [177,178,179]. The glioblastoma growth and proliferation are inhibited when PDK1 and epidermal growth factor receptor (EGFR) are targeted together, reversing the Warburg effect through decreased HIF-1 expression, shifting the Warburg phenotype to OXPHOS, and inhibiting glioblastoma multiforme growth and proliferation [180]. In addition, several dietary and medicinal plants, such as oleanolic acid and curcumin, inhibit the Warburg effect by altering pyruvate kinase (PK), which reduces aerobic glycolysis in cancer cells [181,182]. Overall, inhibiting the Warburg signature at various checkpoints of the glycolysis could be a highly effective strategy for chemotherapy. 

### 3.2. The Mitochondrial Respiratory Chain (mRC) 

The proportional contribution of glycolysis and OXPHOS to ATP production differs substantially between cell types; for example, the typical contribution of OXPHOS to ATP synthesis in normal cells is 80%, but it is less than that in cancer cells [175,183]. Contrary to popular perception, it has been demonstrated that the availability of oxygen (O_2_) in solid tumors is the most important driver of the oxygen consumption rate (OCR), where respiratory ability is not significantly diminished as compared to normal cells [184]. The heterogeneity among the cancer types in OXPHOS contribution could be due to mtDNA content. For example, acute lymphoblastic leukemia, non-Hodgkin lymphoma, endometrial cancer, colorectal cancer, ovarian cancer, prostate cancer, head and neck cancer, lung adenocarcinoma, esophageal squamous cell carcinoma, and thyroid cancer all have higher mtDNA content than normal tissue [185,186]. 

The OXPHOS deficiency has been suspected in mitochondrial-based myopathies due to genetic abnormalities in the RCC (either complex I, complex II, or complex IV) [187,188,189]. Metformin, the first-line medication for type 2 diabetes, has shown to have exceptional therapeutic potential in preventing a range of human cancers, as well as anti-cancer activity in preclinical animal models [190]. In addition, metformin has been linked to mitochondrial toxicity due to its complex I inhibitory action [191,192]. 

### 3.3. Mitochondrial Metabolic Regulation and Drug Resistance

The anticancer drugs that target thymidylate synthase and folate-dependent enzymes disturb metabolic pathways, potentially leading to new cancer treatments [193]. A growing body of research implies that defective or abnormal glycolysis is frequently linked to drug resistance in cancer treatment [194,195]. Glycolysis dysregulation is caused by abnormal expression of glycolysis-related enzymes, including hexokinase 2 (HK2), phosphoglycerate mutase 1 (PGAM1), pyruvate kinase M2 (PKM2), pyruvate dehydrogenase (PDH) complex, and lactate dehydrogenase (LDH), which contributes to tumorigenesis, tumor growth, and tumor therapeutic resistance [196,197,198,199,200]. Therefore, targeting such enzymes could be novel strategies to overcome cancer chemoresistance (Figure 3). For example, hexokinase 2 is a crucial enzyme that aids the progression and resistance of cancer cells by promoting tumor glycolysis [201]. Hexokinase 2 has two functions: (1) up-regulation of HK2 causes enhanced glycolysis rates, and (2) the interaction between VDAC and HKII inhibits apoptosis by inhibiting the creation of mitochondrial permeability transition pores [202]. Table 1 shows a list of drugs that target glycolysis-related and OXPHOS proteins to aid the effects of conventional chemotherapy.

## 4. Autophagy: A Potential Target in Cancer Treatment 

### 4.1. Autophagy 

Autophagy is a self-digestion mechanism that allows cells to sequester internal and extracellular substrates and merge with lysosomes for degradation to maintain cellular homeostasis, biosynthesis, and energy requirements amid nutritional scarcity or metabolic stress [226,238,239]. This is a complex biochemical process that involves multiple proteins, notably the autophagy-related (ATG) proteins, in sequential phases of initiation, nucleation–elongation–maturation, fusion, and degradation (Figure 4) [240]. 

Autophagy can be initiated by a variety of intracellular and extracellular factors, including nutrient deprivation, which depletes total amino acids and serum fasting; oxidative stress, which stimulates autophagy to recycle damaged organelles (e.g., mitochondria) and remove protein aggregates; as well as the target of rapamycin (TOR) inhibitors like rapamycin and commodity channel index-779 (CCI-779) [239,241,242]. The mammalian target of rapamycin complex 1 (mTORC1) suppresses autophagy by inactivating the unc-51-like kinase 1 (ULK1) complex, which includes ULK kinase, ATG13, focal adhesion kinase family interacting protein of 200 kDa (FIP200), and ATG101, in nutrient-rich circumstances [243,244]. The ULK complex is activated by the dissociation of mTORC1, which then promotes autophagy via the class III phosphatidylinositol 3-kinase (PI3KC3) complex as a result of a nutritional deficiency [244,245,246]. The PI3KC3 complex regulates the development of autophagic membranes by recruiting many proteins, including vacuolar protein sorting 34 (VPS34), beclin-1, VPS15, autophagy and beclin-1 regulator 1 (AMBRA1), and ATG14-like (ATG14L), which have been identified as new phagophore regulators [247,248,249,250]. Phagophores work by engulfing senescent cytosolic components and elongating into spherical autophagosomes, which are controlled by two ubiquitin-like conjugation pathways: the ATG12 conjugation system and the ATG8 conjugation system, which are two different types of conjugation systems [251,252,253,254]. The covalent conjugation of ATG12 to the ATG5 protein is catalyzed in one system by the E1- and E2-like activities of ATG7 and ATG10, forming the ATG12–ATG5 complex [255,256,257]. The ATG12–ATG5 complex directly associates with ATG16-like (ATG16L), forming the ATG12–ATG5–ATG16L complex, which has an E3-like function in the second ubiquitin-like conjugation pathway [258,259,260,261]. The oligomerization state of ATG16 was shown to affect the development of the approximately 350-kDa complex and autophagic activity [262]. These findings imply that the ATG12–ATG5 conjugate and ATG16 form a multimeric complex regulated by the ATG16 homo-oligomer, and that the 350-kDa complex is necessary for yeast autophagy. The microtubule-associated proteins 1A/1B light chain 3 (LC3) (or ATG8 in yeast) protein is conjugated to a lipid molecule called phosphatidylethanolamine (PE) in the second system [263,264]. The LC3 is first synthesized as a precursor protein (proLC3), and then ATG4 converts proLC3 to LC3-I, which is then conjugated with phosphatidylethanolamine (PE) phospholipid by ATG7 and ATG3, as well as the ATG12–ATG5–ATG16 complex, to create LC3-II [264,265,266,267]. Most outer membrane proteins, including ATG proteins, are dissociated once the autophagosome is formed and matured. The autophagosome joins the lysosome in the perinuclear area to produce the autolysosome [268]. Finally, breakdown products can be recycled and used to support cell growth.

The primary modulator of autophagy, mTOR, is an evolutionarily conserved serine/threonine kinase that transmits a range of autophagy-stimulating signals [269,270]. There are two types of mTOR complexes—mTOR complex 1 (mTORC1) and mTOR complex 2 (mTORC2)—and these are important nodes in signaling pathways that maintain cellular energy homeostasis by coordinating anabolic and catabolic activities [271,272]. The Ras homolog enriched in the brain (RHEB) is a lysosomal GTPase that interacts directly with mTOR and activates it, which is essential for mTORC1 growth factor activation [273,274]. Activated mTORC1 inhibits the autophagic cascade response by phosphorylating ATG13 and ULK1, and thus suppressing ATG14-containing PI3KC3 activity to inhibit the formation of the pre-initiation ULK complex (ULK1, ATG101-FIP200-ATG13) [275,276].

As part of the autophagosome formation process, the cytosolic microtubule-associated protein light chain 3 (LC3-I) is conjugated with phosphatidylethanolamine to create LC3-phosphatidylethanolamine (LC3-II). As a result, the ratio of LC3-II to LC3-I levels can be utilized as a marker for autophagy activation [277,278]. It has been demonstrated that when HepG2 cells were given a combination of lentiviral shLC3 and epirubicin, their survival rate was much lower than when they were given either agent alone [279]. The findings of this investigation reveal that LC3 plays a critical role in hepatocellular carcinoma chemoresistance, and is a novel therapeutic target for hepatocellular carcinoma. ATG3 is required for autophagy by the E3-like activity of the ATG12-ATG5 conjugate, which is produced by ATG12 activation by ATG7 and conjugation to ATG5 via E2-like ATG10. In colorectal cancer, the c-Myc/miR-27b-3p/ATG10 regulatory axis modulates chemoresistance [280].

### 4.2. The Role of Autophagy in Cancer Treatment 

Autophagy modulation serves as both a tumor suppressant and a tumor promoter in cancer [257,265,281,282]. There has been a lot of interest in inhibiting autophagy for cancer therapy since it was discovered that autophagy is a survival route for tumor cells [283,284]. For example, the lysosomotropic and anti-malarial drug hydroxychloroquine (HCQ), which inhibits lysosome function and thus limits the breakdown of autophagy products, is now being investigated in the clinic trial [283,285]. In addition, human cancers frequently activate different pathways, including mTOR, PI3K, and AKT, and numerous pathway component-specific activation or inhibition are currently under research or in clinical trials [286]. For example, the use of PI3K pathway inhibitors in combination with HCQ or autophagy genetic ablation has resulted in increased tumor regression [285,287,288,289]. In combination with autophagy inhibition, dietary adjustment may help improve cancer therapy. For example, in a mouse melanoma model, leucine deprivation fails to activate autophagy, resulting in tumorigenesis impairment [290,291]. The survival function of autophagy aids tumor cells in the face of increased metabolic demands, a hypoxic microenvironment, or cancer therapy [292]. On the other hand, persistent or excessive autophagy has been demonstrated to increase cell death after treatment with particular chemotherapeutic drugs, either by boosting apoptosis induction or facilitating “autophagic cell death” [293,294,295]. Although the molecular mechanisms by which autophagy mediates its effects on both normal and malignant cells are not fully understood, different signaling pathways have been linked to autophagy upregulation or downregulation [239,296,297,298]. In normal biological conditions, autophagy operates at a basal rate to preserve cellular viability and homeostasis [299]. When autophagy is disrupted by chemotherapy or other environmental stresses, several pathways influence autophagy for cell death (cytotoxic autophagy) or adaptation and survival (cytoprotective autophagy), including the mTOR, phosphoinositide-dependent protein kinase B/AKT (PKB/AKT) pathway, AMP-activated protein kinase (AMPK), tumor suppressor 53 pathway, mitogen-activated protein kinase (MAPK) signaling, FOXO3A-PUMA signaling, and the ER stress pathway (Figure 5) [300,301,302,303,304,305,306,307,308,309,310,311,312]. 

Upon chemotherapy, the growth factors like epidermal growth factor (EGF) and insulin-like growth factor (IGF) activate the growth factor receptors (EGFR, IGFR, etc.), which in turn activate the PI3K-PDK1-Akt signaling cascade [247,248]. The upstream signaling of PI3K/AKT, AMPK, p53 and MAPKs is integrated by mTORC1 to regulate the autophagic pathway [313,314,315,316,317,318,319,320]. The activation of pro-oncogenic class-I PI3K (PI3KC1) by insulin receptor substrate (IRS) proteins, IRS1 and IRS2 catalyzes the conversion of phosphatidylinositol-4,5-bisphosphate (PIP2) to phosphatidylinositol-3,4,5-trisphosphate (PIP3), which binds both AKT and phosphoinositide-dependent protein kinase 1 (PDK1) and triggers PDK1 to phosphorylate AKT at Thr308 [313,314,315,316,317,318,319,320]. Tuberous sclerosis complex 2 (TSC2) is phosphorylated by active Akt, which suppresses the TSC complex, a GTPase-activating protein (GAP) complex composed of TSC1/2. The TSC complex inhibits mTOR by inactivating RHEB [321,322].

The energy-sensing kinase AMPK is a crucial pathway in eukaryotic cells for maintaining energy balance, coordination metabolism, and the regulation of autophagy, apoptosis, epithelial–mesenchymal transition (EMT), etc. [323,324,325]. The AMPK cascade induces autophagy by recognizing the amount of AMP and ATP [326,327,328,329]. The AMPK is also activated by calcium/calmodulin-dependent protein kinase kinase (CaMKK) and liver kinase B1 (LKB1) when they phosphorylate the Thr-172 residue in response to chemotherapy [330,331,332,333,334,335,336,337,338]. Activated AMPK promotes the formation of TSC1/TSC2 complex to inhibit mTOR activity, resulting in autophagy induction [317,339,340,341,342]. The AMPK can directly phosphorylate ULK1 by separating mTORC1 from ULK1 and promote autophagy in the starving condition, whereas mTORC1 can limit AMPK’s activity by phosphorylating ULK1 in a glucose abundant situation [317,339,340,341,342]. 

The p53 tumor suppressor gene regulates autophagy either by nuclear p53 to stimulate autophagy via transcriptional activities or cytoplasmic p53 to restrain autophagy via cytoplasmic functions [343,344,345,346,347,348]. The activated AMPK causes autophagy through the TSC1/TSC2 complex and the mTOR pathway when nuclear p53 production is boosted by external stress [349,350]. It was discovered that p53 increases the expression of its downstream substrates Sestrin, which then activates AMPK, resulting in the deactivation of mTORC1 and the induction of autophagy [351]. Other key downstream substrates of nuclear p53 include the damage regulated autophagy modulator (DRAM) and death-associated protein kinase 1 (DAPK1), both of which have been shown to activate autophagy [352,353,354,355,356]. The DAPK-1 induces autophagy and membrane blebbing via binding to a linear peptide motif in microtubule-associated protein 1B (MAP1B) or directly phosphorylates Beclin1, a necessary component of the Vps34 complex [357,358,359,360]. On the other hand, cytoplasmic p53 suppresses autophagy through blocking AMPK, TIGAR (TP53-induced glycolysis and apoptosis regulator), and Beclin-1 [361,362,363,364,365,366]. 

MAPKs are involved in a variety of biological activities, including cell development, proliferation, and autophagy [367]. MAPK signaling can be triggered by a variety of cell surface growth factor receptors (i.e., HER2/EGFR, and HER2) in response to external stimuli including chemotherapy [368,369,370,371,372]. The activation of extracellular signal regulated kinases (ERK) through the Ras-Raf-MEK-ERK signal pathway can induce autophagy by the blocking of the TSC1/TSC2 complex, leading to the increased expression of Rheb-GTPase to induce mTORC1, which can result in the inhibition of autophagy [373,374,375,376].

Autophagy’s ability to recycle macromolecules gives cancer cells a competitive edge in harsh situations including metabolic stress, glucose deprivation, oxidative damage, hypoxia, expression of aggregate-prone proteins, ER stress, and so on. Among various stressors, ER stress can successfully trigger autophagy because tumor cells must reuse their organelles to maintain growth [377,378]. Autophagy is induced by ER stress via the protein kinase R (PKR)-like endoplasmic reticulum kinase (PERK)/eukaryotic initiation factor 2 (eIF2) and IRE1/JNK1 pathways [379,380,381]. The unfolded-protein response (UPR) is dominated by the phosphorylation of eIF2, translation suppression, and selective translation of the transcription factor (TF) ATF4 [379]. Phosphorylation of PERK/eIF2 has been found to be required for the transcription of critical autophagy-associated genes (ATG) during ER stress, and may also play a role in the formation of LC3-II [308,380]. Melatonin-induced suppression of gastric cancer (GC) cell proliferation is mediated by activation of the IRE/JNK/Beclin1 signaling pathway, according to recent research [382]. JNK pathway activation has been shown to cause selective phosphorylation of c-Jun and up-regulation of c-Fos [383,384]. The JNK also controls autophagy by regulating ATG transcription via FOXO [385,386,387]. 

Through its established roles in genomic stability, elimination of endogenous sources of ROS, maintenance of bioenergetic functions, degradation of oncogenic proteins, and induction of immune response mechanisms against malignant transformations, autophagy has been involved as a desirable pathway for repressing cancer initiation at distinct steps [388,389,390]. For example, genetic studies of autophagic machinery, such as Atg7, Atg5, and Beclin1, have found that when autophagy is disrupted, tumor initiation accelerates [391,392,393,394]. Autophagic cell death (ACD) is a type II cell death process that does not cause real apoptosis, and involves an increase in autophagic flux before or during cell death rather than only an increase in autophagy markers [395,396,397]. Chloroquine has been shown to destroy breast and pancreatic cancer stem cells (CSCs) by inhibiting autophagy via the Janus kinase 2 (JAK2) and DNA methyltransferase 1 (DNMT1) pathways [398,399,400]. However, in another study, autophagy was observed to promote the mortality of CSCs, which is caused by an increase in autophagy-related proteins and autophagosomes in CSCs from malignant gliomas [401,402]. Autophagy induces cell death and regulates cell proliferation by accelerating genetic stability through removing oncogenic proteins [391]. For example, autophagy has been found to destroy a number of proteins implicated in oncogenesis, including the mutant form of p53, DRAM, p62, NF-κB, promyelocytic leukemia/retinoic acid receptor α (PML/RARα), BCR-ABL1, and sequestosome 1 (SQSTM1) [352,403,404,405,406,407,408,409]. In addition, autophagy’s inducement stress-related response is one possible method by which autophagy serves as a tumor suppressor, including oncogenic stress, ER-stress and DNA-damage-stress, etc. [410,411]. Cellular senescence is a program of permanent cell division arrest that can be produced in response to oncogenic stress to avoid malignant transformation [412,413]. Increased levels of ER-stress and DNA-damage-stress resulted from autophagy suppression, which in turn contributed to oncogenesis through increased levels of p62 accumulation [405,414,415,416]. 

In addition to its tumor suppressive role, autophagy supports the tumor microenvironment of cancer cells in stress conditions, including anti-cancer treatment, metabolic stress, oxygen and glucose deprivation, genotoxic, and oxidative stress. This tumor-promoting role of autophagy is achieved through supplying nutrients and energy to cancer cells, mediating adaptation to hypoxia, oxidative stress and DNA damage, supporting stromal cells to maintain homeostasis and tumor growth, and regulating UPR [417]. In the stroma, autophagy’s activation protects nearby cancer cells from cellular damage and death by recycling energy-rich metabolites such as the ketones and L-lactate, leading to improved mitochondrial biogenesis [418]. Autophagy is an important method for cancer cells to cope with UPR and to promote their survival [419]. In the majority of cancer cells, the PERK-arm of the UPR is critical for autophagy’s activation; for example, both phospho-mutant eIF2 and dominant-negative PERK have been found to impede LC3-I to LC3-II conversion [308,379]. 

## 5. Cancer Chemoresistance: Mitochondrial Fusion, Fission, Biogenesis, and Mitophagy 

### 5.1. Mitochondrial Fusion and Fission 

Mitochondrial biogenesis, fusion, fission and mitophagy, collectively referred to as mitochondrial dynamics, are important in maintaining mitochondrial homeostasis in response to variations in energy and stress [420]. The beginning and advancement of various cancer types, as well as cancer metastasis, CSC survival, and drug resistance, have all been linked to defective mitochondrial dynamics (Figure 6) [421,422,423]. Fusion happens when two adjacent mitochondria come together, whereas fission occurs when one mitochondrion splits into two [424,425,426]. These two processes are constantly counterbalanced to maintain mitochondrial homeostasis without interfering with each other’s functions [427]. Mitochondrial fusion and fission have a role in a number of cellular activities, including cell cycle development and apoptosis [428,429]. Fusion is frequently linked to high energy demand, resulting in a hyperfused mitochondrial network that produces more ATP and protects against autophagy [424,430,431,432,433,434]. Fission is mostly associated with apoptosis, facilitating mitophagy, promoting mitotic mitochondrial fragmentation and segregating damaged mitochondrial subdomains for elimination [430,435,436,437]. Notably, fusion and fission can be proapoptotic or antiapoptotic depending on the condition of the cell, confounding their involvement in cancer [438,439,440]. The formation of homotypic and heterotypic oligomers of MFN1 and MFN2, which leads to membrane clustering in a GTP-dependent manner, is primarily responsible for fusion of the OMM, whereas OPA1 requires MFN1 to mediate mitochondrial fusion at the IMM [441,442,443,444,445,446]. Fusion requires the co-existence of long and short OPA1 isoforms (L-OPA1 and S-OPA1), although only the L-OPA1 has been shown to be fusion competent [447,448]. Mitochondrial fission is predominantly regulated by dynamin-related protein 1 (DRP1), a cytosolic guanosine triphosphatase (GTPase), recruited by OMM receptors, such as fission protein homolog 1 (FIS1), mitochondrial fission factor (MFF), and mitochondrial dynamics proteins (MIDs) 49/51 [449,450]. DRP1 oligomerizes and coils around the constriction sites of dividing mitochondria when it reaches the OMM, facilitating mitochondrial fission [451]. 

It has been revealed that chemoresistant ovarian cancer cells had more interconnected mitochondrial networks than their chemosensitive counterparts, implying that mitochondrial fusion plays a role in chemoresistance [452,453,454,455]. The mitochondrial membrane potential collapses early in apoptosis, allowing total cytochrome c release from mitochondria to the cytosol [456]. The fusion GTPase OPA1 has the ability to prevent cytochrome c mobilization and release [457]. It has been demonstrated that venetoclax-resistant AML cells are also resistance to cytochrome c release whenever stimulated due to upregulation of OPA1 [458,459]. Since MFN2 deletion greatly enhances Jurkat doxorubicin sensitivity, MFN2 and OXPHOS have been discovered to be considerably elevated in surviving leukemia cells [460,461]. It has been discovered that suppressing MFN1 improves cisplatin sensitivity in human neuroblastoma cells by inhibiting mitochondrial fusion [462]. Even though mitochondrial fission is the polar opposite of mitochondrial fusion, mounting data suggests that it is similarly important in chemoresistance [463]. For example, the chemosensitivity of nasopharyngeal cancer (NPC) cells to cisplatin was boosted by metformin or cucurbitacin E, which inhibited the DRP1 upstream kinase AMPK or cyclin B1/Cdk1 [464]. Similarly, mitochondrial fission driven by DRP1 phosphorylation enhances cell survival and cisplatin resistance in EBV-LMP1-positive nasopharyngeal cancer [465]. As cancer cells adapt to the hypoxic tumor microenvironment, chemoresistant cells may be able to target mitochondrial fission. For example, inhibition of DRP 1 by either a putative DRP inhibitor (e.g., Mdivi-1) or DRP1 silencing increased the cisplatin sensitivity of hypoxic ovarian cancer cells [455,466]. Mdivi, a Drp1 inhibitor, abolished invasion and chemoresistance induced by LASS2 depletion. The mesenchymal stem cells (MSCs)-induced drug resistance is linked to ERK/Drp1-dependent mitochondrial fission [467]. PD98059, an ERK inhibitor, inhibits LASS2-deficient bladder cancer invasion and chemoresistance by suppression of Drp1 phosphorylation [468,469]. Anticancer drugs that increase the release of the high-mobility group box 1 protein (HMGB1) trigger ERK1/2-mediated DRP1 phosphorylation in colorectal cancers, or increased ROS production and hypoxia in ovarian tumors, and are usually associated with fission-driven chemoresistance [470]. Conversely, induction of DRP1 through inhibiting anti-apoptotic proteins BCL-2 or activating Sirt3 (histone deacetylase) by ABT737 (an inhibitor of antiapoptotic BCL-2/BCL-X) promotes fission, leading to apoptosis and mitophagy in ovarian cancer cells resistant to cisplatin [453,455,471]. 

### 5.2. Mitochondrial Biogenesis and Mitochondrial Selective Autophagy (Mitophagy)

Mitochondrial biogenesis and mitochondrial selective autophagy (mitophagy), two opposing cellular processes, coordinately control mitochondrial content to support energy metabolism, in response to the cellular metabolic conditions, stress, and other intracellular or external signals [472]. Mitochondrial biogenesis is a multistep biological process that includes mtDNA transcription and translation, as well as the translation of nucleus-derived transcripts [473]. The recruitment of newly synthesized proteins and lipids, import and assembly of mitochondrial and nuclear products in the expanding mitochondrial network are the principal roles of mitochondrial biogenesis. The peroxisome proliferator-activated receptor-gamma co-activator 1-alpha (PGC-1α), the well-studied member of the PGC family, coordinates the activity of many transcription factors involved in mitochondrial biogenesis and function [474]. For example, PGC-1 overexpression improves mitochondrial function and promotes mitochondrial proliferation in mouse models [475]. Furthermore, the expression of PGC-1, and/or its homolog PGC-1, promotes mitochondrial respiration in complex III or IV impairment. As a result, PGC-1 has been dubbed the “master regulator” of mitochondrial biosynthesis and function. In addition, PGC1-expressing cells have a better chance of surviving against metabolic stressors such as oxidative damage, energy deprivation, and even cancer therapy [476]. Based on the evidence, PGC1 is thought to play a function in chemotherapy resistance. The most common techniques for disrupting PGC1 signaling have been to block critical enzymes in PGC1-dependent metabolic pathways or to target PGC1-related transcription factors as PGC1 inhibitors are not yet available. For instance, inhibitors of OXPHOS, FAO, or ERR all inhibited cell proliferation in PGC1-positive cancer cells [477]. Suppressing PGC1 or OXPHOS expression in melanoma cells re-sensitizes therapeutic-resistant cells to oxidative damage [478]. The PGC1 suppression increases ROS generation, which stabilizes HIF1 protein and causes a metabolic transition from OXPHOS to glycolysis, which enhances cell survival. It has been reported that triple suppression of PGC1α, HIF1α, and glutamine utilization results in complete blockage of tumor growth [479]. The PGC1 was found to be a tumor suppressor in several cancer types in a few investigations [480,481,482]. Low PGC1 expression, for example, has been linked to poor prognosis in VHL-deficient clear cell renal cell carcinoma (ccRCC) and breast cancer patients [480]. PGC1 increases Bax-mediated apoptosis in colorectal and ovarian epithelial carcinoma cells and works as a stabilizer of the tumor suppressor mitostatin, which triggers mitophagy in breast cancer [482,483,484]. 

Mitophagy is the process of selective removal of damaged mitochondria by the autophagic machinery. In many mammalian cell types, mitochondrial fission is shown to be synchronized with mitophagy, implying that it is a precondition for the mitochondrial breakdown process [485]. The mitophagy induction process begins with the engulfment of damaged mitochondrial components by an isolating membrane (phagophore). Mitophagy involves vesicle nucleation, vesicle elongation, and retrieval, which is followed by the creation of a mitophagosome, a double-membrane structure that sequesters the organelle and allows mitophagosomal destruction after fusion with a lysosome [486,487]. During the degradation process, amino acids and fatty acids are produced, which can be used for protein synthesis or oxidized by the mitochondrial ETC to produce ATP that can be used for cell survival [488]. Essential components for cell viability (e.g., mitochondria) can be degraded with a lethal process, leading to autophagic cell death [489,490,491,492]. Many studies have found that the mitophagy-related molecules phosphatase and tensin (PTEN)-induced kinase 1 (PINK1), parkin, Bcl-2/adenovirus E1B 19 kDa interacting protein 3 (BNIP3), FK506-binding protein 8 (FKBP8) and FUN14 domain-containing protein 1 (FUNDC1) promote tumor formation and can be used as therapeutic targets in a variety of malignancies [493,494,495,496]. For example, PINK1 overexpression is linked to a poor response to chemotherapy and a poor prognosis in patients with esophageal squamous cell carcinoma treated with neoadjuvant chemotherapy, while silencing PINK1 inhibits lung cancer cell growth and migration while also inducing cell death [495,497]. In addition, polyphyllin I, a natural compound, decreased tumor growth and promoted apoptosis in MDA-MB-231 xenografts, and PINK1 knockdown boosted these effects [498]. Furthermore, FUNDC1 overexpression predicts a poor prognosis and is a possible target for improving chemoradiotherapy effects in cervical cancer patients. In contrast, a study found that the migraine medicine dihydroergotamine tartrate (DHE) could cause lung cancer cell death via mitophagy and mitochondria-dependent cell apoptosis through the activation of the PINK1/Parkin pathway [499]. In addition, translocation of BNIP3 to the mitochondria causes cytochrome c to be released, resulting in caspase activation and apoptosis in cardiomyocytes [500]. Arsenic trioxide, ceramide, and concanavalin A can stimulate BNIP3, and BNIP3 is involved in the induction of autophagy and cell death in response to these stimuli [501,502,503].

Drug resistance frequently results in treatment failure due to autophagy or mitophagy. Mitophagy has a dual role in cancer therapy: on the one hand, it can cause cancer cell death, and on the other, it can help cancer cells survive [504]. Importantly, drug resistance to major chemotherapy medications, including cisplatin, doxorubicin, 5-fluorouracil, and paclitaxel, typically results in treatment failure due to autophagy or mitophagy [505]. The PINK1/Parkin pathway is the well-studied ubiquitin-mediated system involved in cancer chemoresistance. PINK1 aggregation at the OMM is triggered by mitochondrial malfunction or depolarization, which recruits and phosphorylates Parkin, an E3 ligase, boosting its E3 ligase activity [506,507,508,509,510]. B5G1, a novel betulinic acid analog, sensitizes multidrug-resistant cancer cells via inhibiting PINK1/ Parkin -dependent mitophagy [511]. Several mitophagy receptors, including BNIP3, BNIP3-like (BNIP3L, also known as NIX)), and FUNDC1, have been linked to chemoresistance in PINK/ Parkin -independent mitophagy [512,513,514,515,516]. Mitophagy produced by doxorubicin leads to drug resistance, and BNIP3L silencing improves doxorubicin sensitivity in human colorectal CSCs isolated from HCT8 cells, implying that mitophagy plays a role in drug resistance [517]. FUNDC1 overexpression stimulates the development of cervical cancer cells, while inhibition of its expression improves susceptibility to cisplatin and ionizing radiation [518]. 

### 5.3. Autophagy Acts as a Chemotherapy Resistance Machinery in the Mitochondrial Pathway

The main principles of cancer therapy are the induction of cell death and the inhibition of cell survival. According to an increasing body of research, autophagy contributes to the anticancer efficiency of chemotherapy by inducing the apoptotic pathway, but it also confers drug resistance via its paradoxical protective role in many circumstances (Figure 7) [519]. Recent research shows that increased autophagy not only improves tumor survival, but also improves tumor drug resistance in a variety of tumor types. Autophagy influences both the efficacy of tumor chemotherapy and the degree of drug resistance, and this is likely because cancer cells use mitophagy to rapidly sweep away damaged mitochondria in order to mediate their own drug resistance [497,520]. According to several studies, the ATG has a crucial role in drug resistance mediated by mitochondria. For example, ATG12 conjugation regulates mitochondrial homeostasis and cell death, whereas altering the ATG12–ATG3 complex leads to increased mitochondrial mass, fragmentation of the mitochondrial network, and resistance to cell death mediated by mitochondrial pathways [521]. It has been demonstrated that miR-1 overexpression increased the cisplatin sensitivity by suppressing the ATG3-related autophagy of NSCLC cells [522]. Similarly, by inhibiting ATG5 expression and autophagy, miRNA-153-3p increases gefitinib sensitivity in NSCLC. Recently, ATG5 has been found to be implicated in doxorubicin resistance in gall bladder cancer [523]. It has been shown that knocking down ATG5 in paclitaxel-resistant cells causes a G2/M arrest and makes cells susceptible to paclitaxel-induced necrosis [524]. Another study showed that direct knockdown of ATG5 increased paclitaxel resistance in Ras-NIH 3T3 (v-Ha-ras-transformed NIH 3T3) cells by reducing the frequency of early apoptotic cells [525]. Direct control of beclin-1 (ATG6) is also critical in the development of drug resistance. For example, in ER (estrogen receptor)-positive breast cancer cells, suppressing beclin-1 boosted tamoxifen sensitivity in vitro, and lower beclin-1 expression predicts a better prognosis in patients with ER-positive breast cancer [526]. MicroRNA-30a downregulation enhanced osteosarcoma cell chemoresistance by activating beclin-1-mediated autophagy, which decreased cell proliferation and invasion [527]. In addition, miR-30a has been shown to sensitize tumor cells to cisplatin by lowering beclin-1-mediated autophagy, suggesting that boosting miR-30a levels in tumor cells could be a novel way to improve chemotherapy’s efficacy during cancer treatment [528]. In MCF-7 cells, co-treatment with docetaxel and siATG7 resulted in a higher cytotoxicity and apoptosis rate than docetaxel treatment alone, demonstrating that silencing of ATG7 increased the effectiveness and apoptotic effect of docetaxel [529]. ATG12 is important for autophagy, and studies have shown that miR-23b-3p can suppress autophagy in gastric cancer cells, restoring the proliferation of chemoresistant cells in vitro and in vivo [530]. Through Nrf2-dependent metabolic reprogramming, p62/SQSTM1 increases the malignancy of HCV-positive hepatocellular carcinoma [531]. The abnormal amplification and phosphorylation of p62/SQSTM1 has been linked to tumor growth and cisplatin resistance in patient-derived high-grade serous ovarian cancer cells [532]. 

Several upstream target genes and signaling pathways have been shown to influence autophagy-mediated chemoresistance, in addition to directly influencing ATG as indicated above, including DNA-dependent protein kinase catalytic subunit (DNA-PKcs), high mobility group box 1 (HMGB1), high-mobility group nucleosome-binding domain 5 (HMGN5), heat shock protein 90AA1 (HSP90AA1), insulin growth factor 2 (IGF2), and N-myc downstream regulated gene 1 (NDRG1) [533,534,535,536,537]. The induction of autophagy by oxidative stress is thought to contribute to the long-term survival of breast cancer cells by regulating DNA repair via ataxia telangiectasia mutated (ATM), DNA-PKcs, and poly (ADP-ribose) polymerase (PARP)-1 [538,539]. HMGB1 has become a new target for chemotherapy because of its autophagy regulation in response to oxidative stress [540,541]. HMGB1 is a prototypical damage-associated molecular pattern (DAMP) molecule that is released by induced autophagy and enhances treatment resistance in ovarian cancer, colorectal cancer, and lung cancer [470,542]. The release of the HMGB1 and the receptor for the advanced glycation end-product (RAGE) triggers autophagy, which leads to chemoresistance and regeneration of cancer cells [543,544]. The use of an HMGB1 inhibitor or a RAGE blocker to improve chemotherapeutic sensitivity in colorectal cancer is well-accepted [470]. HMGN5 is a novel target for enhancing osteosarcoma therapy because it regulates autophagy, a key component in the development of chemoresistance [535]. HSP90AA1 increases chemotherapy (doxorubicin, cisplatin, and methotrexate) resistance in osteosarcoma through upregulation autophagy [534]. The pro-survival action of HSP90AA1 could be reversed by 3-methyladenine (3-MA) through inhibiting HSP90AA1 and increased chemosensitivity via blocking autophagy. In human osteosarcoma cell lines, IGF1R-targeted treatment improved doxorubicin chemosensitivity [536]. Activation of IGF or insulin signaling has been shown to prolong the lifespan of osteosarcoma cells under chemotherapeutic stress, resulting in a drug-resistance [545]. The anti-metastatic activity of cells expressing NDRG1 makes this protein an important therapeutic target for cancer chemotherapy [537]. NDRG1 inhibits both basal and hypoxia-induced autophagy at the initiation and degradation stages, as well as making pancreatic cancer cells more susceptible to lysosomal membrane permeabilization [546]. 

The Ca^2+^ is an important intracellular second messenger that regulates significant cell fate decisions like metabolism, growth, and death [547]. Mitochondria are considered as a crucial cell death checkpoint, and mitochondrial Ca^2+^ excess is thought to be a powerful inducer of apoptosis through the intrinsic pathway. One of the ways to trigger mitochondrial-mediated intrinsic apoptotic pathway is mitochondrial Ca^2+^ overload, which is caused by a persistent rise of calcium ion [24,548,549]. According to a growing number of studies, Ca^2+^ is also involved in the latest hallmarks of cancer like autophagy, which has been linked to tumor growth and cancer therapy [25,292,550]. As a result, targeting mitochondria, specifically mitochondrial Ca^2+^ homeostasis and autophagy, is a main focus of anti-cancer therapy in order to sensitize cancer cells to death and overcome drug resistance [551]. 

Uncontrolled Ca^2+^ dynamics affect practically every aspect of cell function, including proliferation, gene expression, cell death, and protein phosphorylation and dephosphorylation due to the complicated network of Ca^2+^ transporters or pumps is disrupted [552,553,554]. The downregulation of calcium release-activated calcium modulator 1 (CRACM1, also called Orai1) has been demonstrated to delay cytoplasmic Ca^2+^ clearance, boosting the activation of many Ca^2+^-dependent kinases and enzymes, leading to activation of the cyclin-dependent kinase inhibitor p21, which results in autophagy activation, cell growth arrest, and enhanced cell survival [555]. Inhibition of ORAI1-mediated Ca^2+^ entry, on the other hand, increases the chemosensitivity of HepG2 hepatocarcinoma cells to 5-fluorouracil [556]. Furthermore, several Ca^2+^-mobilizing substances (thapsigargin, ATP, ionomycin, and chemotherapeutic drugs) and nutritional deprivation cause increases in Ca^2+^ levels, as well as the activation of pro-survival autophagy in cancer cells [557,558,559]. 

The inner mitochondrial membrane of diverse tissues has been found to contain the following types of potassium channels: ATP-sensitive, Ca^2+^-activated, voltage-gated, and two-pore domain potassium channels [557,560,561,562,563,564,565,566]. The regulation of mitochondrial respiration, membrane potential, and ROS production are all direct functions of these channels [567]. The mitochondrial potassium channel act as important factors in various pro-life and pro-death reactions that are triggered by changes in these channel activities in distinct cell types. In response to a change in mitochondrial Ca^2+^ concentration, Ca^2+^ regulates three types of mitochondrial potassium channels: mitoBKCa, mitoIKCa, and mitoSKCa [567,568]. Because of their putative role in cytoprotective events, potassium channels found in heart or brain mitochondria have received a lot of interest [567,568,569,570,571,572,573]. For example, in cardiac and neural tissue, activation of mitochondrial potassium channels, such as ATP-regulated or calcium-activated large conductance potassium channels may have cytoprotective effects [574]. It is critical to design a therapy that encourages cancer cells to undergo apoptosis in spite of such limitations, like Bax or Bak, which causes apoptosis resistance and limits the activity of chemotherapeutics. The Bax protein blocks the mitochondrial Kv1.3 potassium channel, making it a potential therapeutic target [575]. For example, three inhibitors (Psora-4, PAP-1, and clofazimine) of mitochondrial Kv1.3 channels cause cancer cells to die in a Bax/Bak-independent manner [576].

## 6. Conclusions

Overall, this work provides strong evidence for the roles of autophagy and mitochondria in cancer chemoresistance. In many malignancies, autophagy modulation serves as both a tumor suppressor and a tumor promoter. Autophagy helps tumor cells survive and confers drug resistance in numerous situations when they are subjected to increased metabolic demands during cancer therapy. Survivability of cancer cells in the presence of abnormal glucose metabolism, fatty acid synthesis, and glutamine metabolic rates is linked to glycolysis and OXPHOS, which is important not only for carcinogenesis, but also for chemoresistance and recurrence, and especially for the emergence of treatment failure. Chemotherapy-induced mitophagy aids mitochondrial biogenesis by allowing damaged proteins or organelles to be recycled, preventing DNA damage and enhancing drug resistance. As a result, discovering autophagy inhibitors or activators, as well as targeting glycolysis and OXPHOS, could be critical in reversing drug resistance and enhancing chemosensitivity. However, combining autophagy and glycolysis, as well as OXPHOS modulators and chemotherapeutic medicines, will offer new hope for cancer treatment.

## Figures and Tables

**Figure 1 cancers-14-01462-f001:**
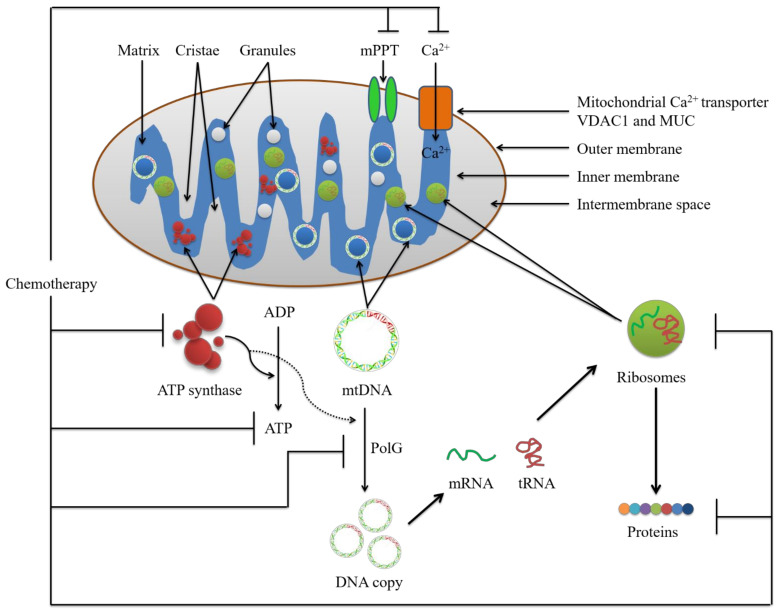
Chemotherapy in development targeting mitochondrial components. The mitochondrial components are being targeted by chemotherapeutics that regulate metabolic activity, enabling for the selective killing of cancer cells for therapeutic benefit. To achieve therapeutic benefit, the calcium ion (Ca^2+^) channels or other ion complexes are predominantly targeted by chemotherapy, such as the voltage-dependent anion channel 1 (VDAC1), the mitochondrial calcium uniporter (MCU) complex, and the mitochondrial permeability transition pore complex (mPTPC), etc. Chemotherapy also targeted various mitochondrial enzymes or proteins, including mitochondrial ribosome and ribosomal proteins (MRP), DNA polymerase subunit gamma (PolG) and ATP synthase, etc. The mitochondrial deoxyribonucleic acid (mtDNA) and mitochondria-derived reactive oxygen species (mtROS) are considered to be potential mitochondrial components for chemotherapy. mPPT: mitochondrial permeability transition pore complex; ADP: adenosine diphosphate; ATP: adenosine triphosphate; mtDNA: mitochondrial deoxyribonucleic acid (DNA); mRNA: messenger ribonucleic acid; tRNA: transfer ribonucleic acid; VDAC1: voltage-dependent anion channel1; MCU: mitochondrial calcium uniporter; PolG: DNA polymerase gamma subunit.

**Figure 2 cancers-14-01462-f002:**
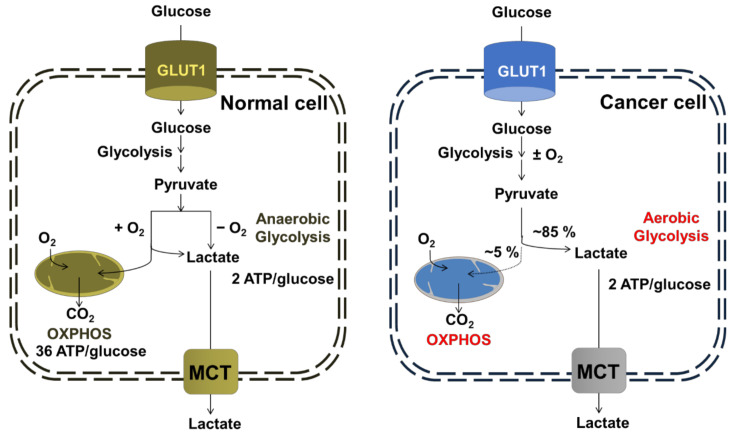
Metabolic activity of normal and cancer cells. The cells use glucose as their primary metabolic substrate, which is broken down into pyruvate by glycolytic enzymes, which ultimately produces adenosine triphosphate (ATP) either by glycolysis or oxidative phosphorylation (OXPHOS). In normal cells, most pyruvate is transported into mitochondria for further oxidation to produce ATP. In cancer cells, most pyruvate is converted to lactate and transported out through the MCT transporter. GLUT: glucose transporter; MCT: monocarboxylate transporters.

**Figure 3 cancers-14-01462-f003:**
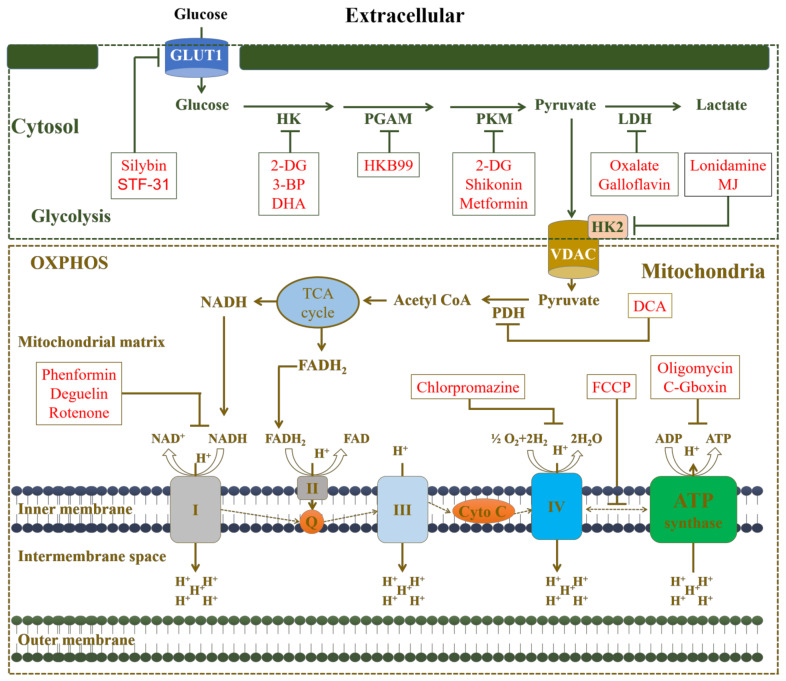
Drugs targeting glycolysis and oxidative phosphorylation (OXPHOS) to sensitize chemotherapy resistance cancer cells. The cells use glucose as their primary metabolic substrate, which is broken down into pyruvate by glycolysis, which produces adenosine triphosphate (ATP), and nicotinamide adenine dinucleotide (NAD) plus hydrogen (H) (NADH) in the presence of various glycolysis enzymes, including hexokinase (HK), phosphofructokinase (PFK), and phosphoglycerate mutase (PGAM), etc. In normal cells, most pyruvate is transported into mitochondria for further oxidation, whereas in cancer cells most pyruvate is converted to lactate in the presence of lactate dehydrogenase (LDH). The tricarboxylic acid (TCA) cycle is fueled by acetyl-CoA, which produces NADH, flavin adenine dinucleotide (FADH_2_), ATP, and metabolic intermediates required for proliferation. The mitochondrial respiratory chain complex (RCC) or electron transport chains (ETC) are composed of complex I, complex II, complex III, complex IV and complex V (ATP synthase), that catalyzes the oxidation of reducing equivalents, primarily NADH and FADH_2_, in the IMM using the terminal electron acceptor oxygen (O_2_). The ETC receives NADH and FADH_2_ from the TCA cycle in the mitochondrial matrix, each of which provides a pair of electrons to the ETC via complexes I and II, respectively. The ETC complexes (complexes I–IV), the ATP synthase, the IMM, two diffusible electron carriers, and substrate transporters in IMM form the OXPHOS system. Ubiquinone (Q), which can be reduced to ubiquinol (QH_2_), is the product of electron transfer from complex I or II. Complex III oxidizes ubiquinol, allowing one electron to continue its journey via cytochrome c at a time. Cytochrome c transfers electrons to complex IV, where they are reduced to water by acting as a terminal electron acceptor. The reduction of O_2_ to H_2_O results in the pumping of protons (H^+^) to the IMS at complex IV. The re-entry of H+ into the matrix via complex V is related to the generation of ATP from ADP. OXPHOS: oxidative phosphorylation; GLUT1: glucose transporter 1; NADH: nicotinamide adenine dinucleotide (NAD) hydrogen; HK: hexokinase; PFK: phosphofructokinase; PGAM: phosphoglycerate mutase; LDH: lactate dehydrogenase; TCA: tricarboxylic acid; FADH_2_: adenine dinucleotide; RCC: respiratory chain complex; 2-DG: 2-deoxyglucose; 3-BP: 3-bromopyruvate; DHA: dehydroascorbic acid; MJ: methyl jasmonate; PKM: Pyruvate kinase M; PDH: pyruvate dehydrogenase; DCA: dichloroacetate; FCCP: carbonyl cyanide-p-trifluoromethoxyphenylhydrazone.

**Figure 4 cancers-14-01462-f004:**
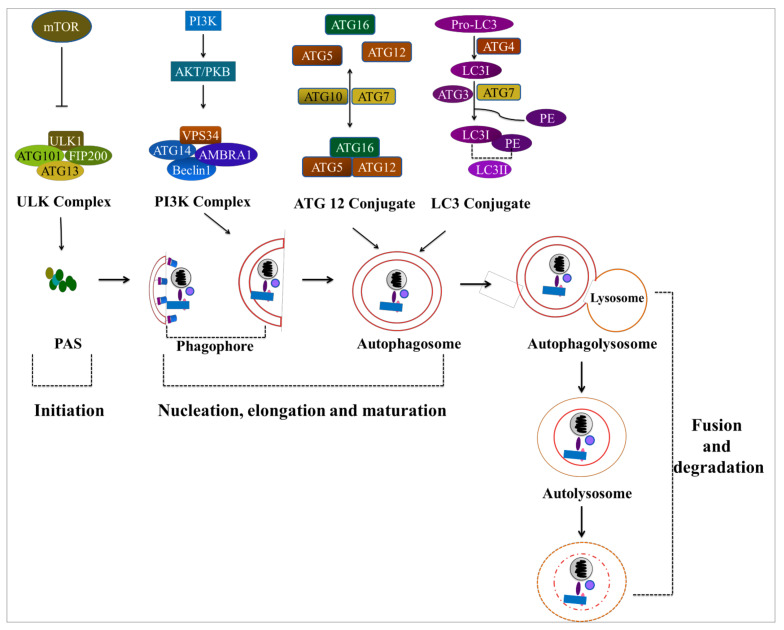
The process of autophagy. The process of autophagy is divided into three major stages: (i) initiation (formation of phagophore assembly site (PAS); (ii) nucleation, elongation and maturation (phagophore regulation and elongating into spherical autophagosomes); and (iii) fusion and degradation (autophagolysosome and autolysosome). mTOR: mammalian target of rapamycin; ULK1: unc-51-like kinase 1; ATGs: autophagy genes; FIP 200: focal adhesion kinase family interacting protein of 200 kDa; PI3K: phosphatidylinositol 3-kinase; PKB: phosphoinositide-dependent protein kinase B; VPS34: vacuolar protein sorting 34; AMBRA1: autophagy and beclin-1 regulator 1; LC3 I/II: light chain I and II; PE: phosphatidylethanolamine.

**Figure 5 cancers-14-01462-f005:**
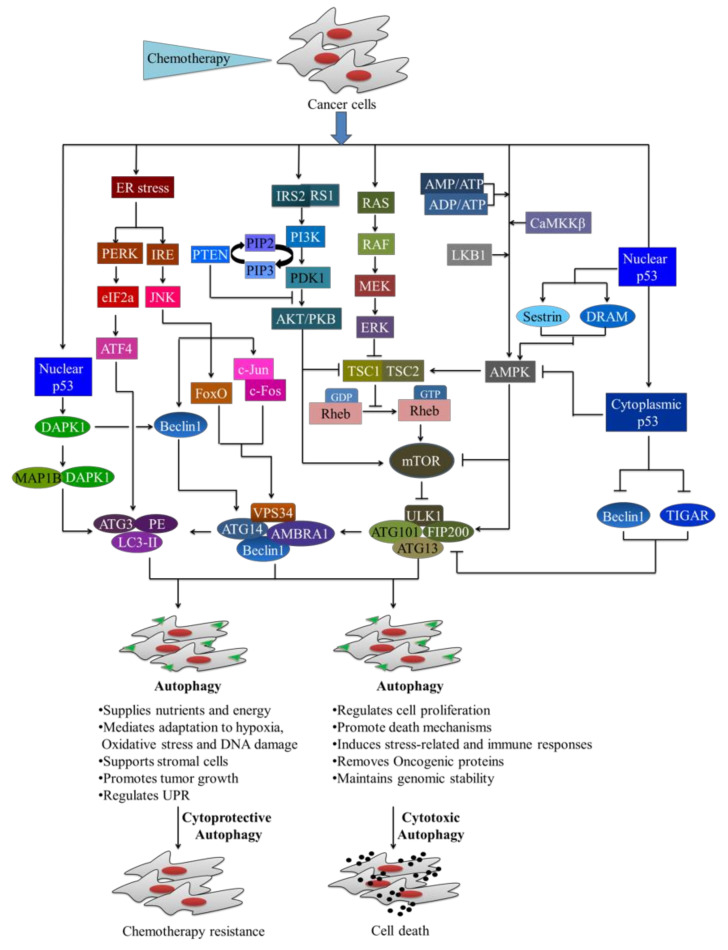
Possible pathways involved in autophagy-mediated cell survival or cell death. Chemotherapy can be activated cytotoxic or cytoprotective autophagy by a variety of signaling pathways; including the mTOR, phosphoinositide-dependent protein kinase B/AKT (PKB/AKT) pathway, AMP-activated protein kinase (AMPK), tumor suppressor p53 pathway, mitogen-activated protein kinase (MAPK) signaling, FOXO3A-PUMA signaling, and ER stress, etc. IRS1/2: insulin receptor substrate 1 and 2; PI3K; phosphatidylinositol 3-kinase; PDK1: phosphoinositide-dependent protein kinase 1; PTEN: phosphatase and tensin; PIP2: phosphatidylinositol-4,5-bisphosphate; PIP3: phosphatidylinositol-3,4,5-trisphosphate; PKB: phosphoinositide-dependent protein kinase B; RAS: rat sarcoma; RAF: rapidly accelerated fibrosarcoma; ERK: extracellular signal regulated kinases; MEK: mitogen-activated protein kinase/ERK kinase; TSC1/2: tuberous sclerosis complex 1 and 2; GDP: guanosine diphosphate; GT: guanosine triphosphate; mTOR: mammalian target of rapamycin; LC3-I/II: light chain 3 I and II; PERK: protein kinase R (PKR)-like endoplasmic reticulum kinase (PERK); eIF2: eukaryotic initiation factor 2; DAPK1: death-associated protein kinase 1; DRAM: damage regulated autophagy modulator; TIGAR: TP53-induced glycolysis and apoptosis regulator; LKB1: liver kinase B1; CaMKK: calcium/calmodulin-dependent protein kinase kinase; Rheb: ras homolog enriched in the brain; AMPK: AMP-activated protein kinase; ATG: autophagy gene.

**Figure 6 cancers-14-01462-f006:**
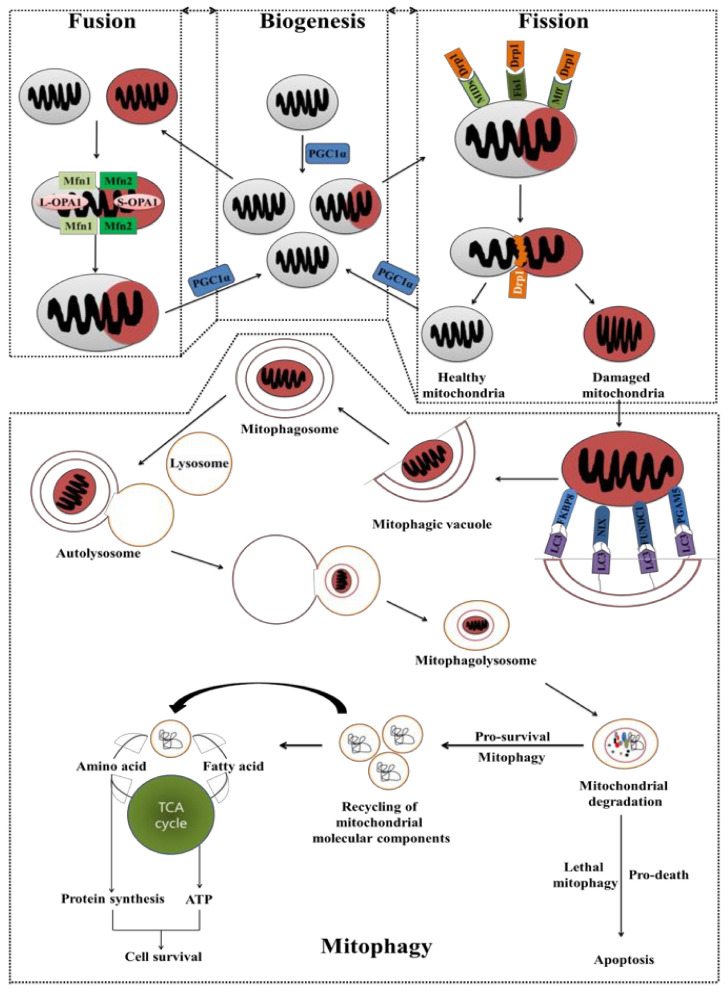
The role of mitochondrial biogenesis and mitophagy in chemotherapy resistance. Mitochondrial biogenesis, fusion, fission, and mitophagy are cellular mechanisms that coordinately control mitochondrial content to maintain energy metabolism in response to cellular metabolic circumstances, stress, and other intracellular or external signals. The mitochondria increase their population by biogenesis process through utilizing fusion to fuse two neighboring mitochondria and fission to split two mitochondria from one, whereas the mitophagy (selective autophagic machinery) selectively removes damaged mitochondria to maintain cellular homeostasis. The pro-survival process of mitophagy utilized recycled mitochondrial components produced during the degradation process to use amino acids and fatty acids for protein synthesis and ATP generation, followed by cell survival (chemotherapy resistance). PGC1α: proliferator-activated receptor-gamma co-activator 1-alpha; OPA1: optic atrophy 1; Mnf1/2: mitofusin 1 and 2; Drp1: dynamin-related protein 1; FIS1: fission protein homolog 1; MFF: mitochondrial fission factor; MIDs: mitochondrial dynamics proteins; PINK1: PTEN-induced kinase 1; BNIP3: Bcl-2/adenovirus E1B 19 kDa interacting protein 3; FKBP8: FK506-binding protein 8; FUNDC1: FUN14 domain-containing protein 1; TCA: tricarboxylic acid.

**Figure 7 cancers-14-01462-f007:**
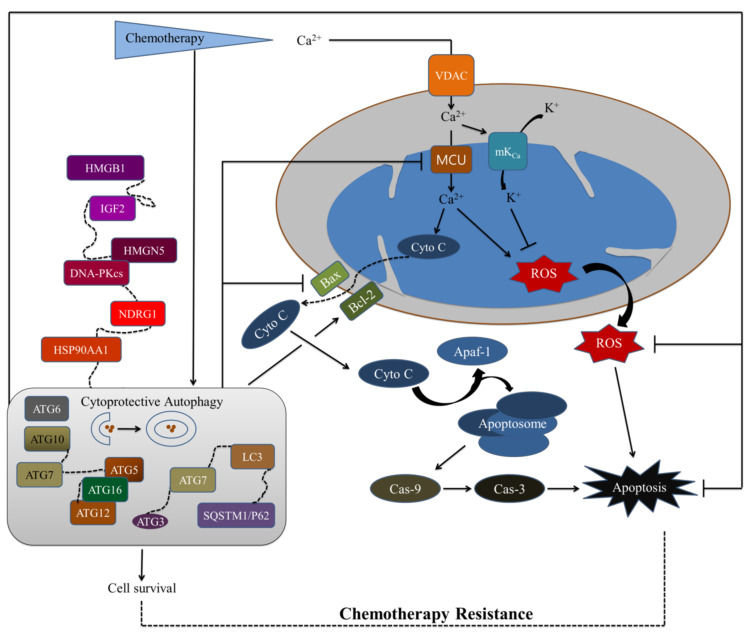
Chemotherapy resistance, mitochondrial dysfunction and autophagic machinery. Mitochondria act as a powerful inducer of apoptosis via the intrinsic pathway, while mitochondrial calcium ion (Ca^2+^) regulates significant cell fate decisions like metabolism, growth, and death. The cytoprotective autophagy activated by chemotherapy through several autophagy regulating proteins that can effectively blocked Ca^2+^ channels on the outer mitochondrial membrane (OMM) and/or intramitochondrial membrane (IMM) to enter Ca^2+^. Inhibition of mitochondrial Ca^2+^ signaling can block the release of mitochondrial cytochrome c (Cyto C), followed by suppression of interaction between Cyto C and apoptosome, subsequently inhibit the apoptosis pathway. In addition, Ca^2+^ and Ca^2+^-activated potassium channel (mKca) has the ability to block the apoptosis pathway through the regulation of reactive oxygen species (ROS). Ca^2+^: calcium ion; HMGB1: high mobility group box 1; HMGN5: high-mobility group nucleosome-binding domain 5; HSP90AA1: heat shock protein 90AA1; IGF2: insulin growth factor 2; NDRG1: N-myc down-stream regulated gene 1; DNA-PKcs: DNA-dependent protein kinase catalytic subunit; mKca: Ca^2+^-activated potassium channel; Cyto C: cytochrome c; Bcl-2: B-cell lymphoma 2; Bax: BCL2-associated X; Apaf-1: apoptotic protease activating factor 1; Cas-9: caspase-9; Cas-3: caspase-3.

**Table 1 cancers-14-01462-t001:** The drugs targeting glycolysis and oxidative phosphorylation (OXPHOS) to overcome different chemotherapy resistance.

Targets	Drugs	Augment the Effects of Chemotherapy	Refs.
GLUT1	Silybin	Doxorubicin, cisplatin and paclitaxel	[203,204,205]
	STF-31	Gemcitabine, paclitaxel and 5-fluorouracil	[156]
Hexokinase (HK)	2-deoxyglucose (2-DG)	Cisplatin	[206]
		Doxorubicin	[207]
		Etoposide	[208]
	Lonidamine	Cisplatin and paclitaxel	[209]
		Carboplatin	[210]
		Doxorubicin	[150,151,211,212,213]
	3-bromopyruvate (3-BP)	5-fluorouracil	[214]
		Tamoxifen	[215]
		Daunorubicin	[216]
	Dehydroascorbic acid (DHA)	Cisplatin or sorafenib	[202,217]
HK2—VDAC complexes	Methyl jasmonate (MJ)	Cisplatin and doxorubicin	[218]
Phosphoglycerate mutase 1 (PGAM1)	HKB99	Erlotinib	[219]
Pyruvate kinase M2 (PKM2)	2-DG	Doxorubicin	[220]
	Shikonin	Tamoxifen	[221]
	Metformin	Cisplatin, doxorubicin and 5-fluorouracil	[222,223]
Pyruvate dehydrogenase (PDH) complex	Dichloroacetate (DCA)	Tamoxifen	[224,225]
		Cetuximab	[226]
Lactate dehydrogenase (LDH)	Oxalate	Taxol	[227]
	Galloflavin	4-hydroxy-tamoxifen	[167,228]
Aldehyde dehydrogenase (ALDH)	Gossypol	Irinotecan	[229,230]
Complex I	Phenformin	Gemcitabine and irinotecan	[229,230,231]
	Deguelin	Vemurafenib	[232]
	Rotenone	Doxorubicin	[233]
Complex IV	Chlorpromazine	Temozolomide	[234]
ATP synthase	Oligomycin	Doxorubicin	[233,235]
	C-Gboxin	Gboxin	[236]
Mitochondrial uncoupler	FCCP	Doxorubicin	[237]

## Data Availability

The data presented in this study are available in this article Cancers.

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
