# Peer review of "Chemotherapy Resistance: Role of Mitochondrial and Autophagic Components"

_cancers, 2022, doi:10.3390/cancers14061462_

Round 1

Reviewer 1 Report

No further comments, authors addressed key concerns 

Author Response

Thank you very much for your valuable comments and suggestions. 

Reviewer 2 Report

Dear Editor,

The authors of the above manuscript have reviewed the recent researches on mitochondria and autophagy in the contest of resistance to chemotherapies. The review is very informative and well-constructed but I would like to suggest some revisions before acceptance for publication.

  1. Check English spelling and typos throughout the text:

Lane 35 I would suggest to start the sentence in a different way than “occurring through”;

Lane 91 (;);

Lane 67, 68 (different font in ref number, also the period is cryptic and needs to be adjusted or splitted in sentences);

Lane 102 (space and paragraph title); lane156 to 158 is a repetition of above sentence; Lane 200-202 is a repetition of above sentence and needs to be explained more the mechanism of action of cisplatin in the context of mt-DNA damage;

Paragraph 2.5 check references and substitute PMID numbers, also the first sentences are quite cryptic and I would suggest to adjust them and explain better the implication of PolG1 tumorigenesis and as a therapy;

Lane 229-230 role of RPS4X has to be explained better;

Lane 304 I would use “drugs” rather than “medicines”. Also, I would check the entire paragraph in terms of English phrase construction and clarify in which types of tumors inhibition of OXPHOS can be used as a treatment and explain better how Warburg effect can be reversed and used for chemotherapy;

Lane 315 adjusts the sentence;

Lane 351 which anticancer drugs?;

Lane 472 Figure 4 is repeated twice;

Lane 502/570 reference needs to be adjusted PMID deleted;

Paragraph 5.1 I would suggest to avoid repetition and make the paragraph more concise and focused on the mitochondria fusion and fission. Also, I would describe more how mitochondria dynamics has been implicated in chemoresistance. In addition, several PMID references have to be corrected;

Lane 767-768 sentence is not complete, check PMID typos in the paragraph. Furthermore, I suggest to add more data on the mitochondrial potassium channels and their implication in cytoprotection;

  1. Add appropriate references to the sentences referred to the following lanes: 42, 43,51,54,82,93,96,101,143,151,153,164,170,197,263,266,272,283,328,455,554,774,776,800
  2. Figure 1: I would explain the figure legend and the mitochondrial components;
  3. Figure 3: It is worth adding mechanism of action of drugs (listed in table 1) on each component of the glycolysis pathway. I would suggest to make a parallel figure or to change Figure 3 to add a cancer cell in addition to a normal cell;
  4. Figure 5: Figure legend has to be adjusted in terms of English using a more formal language;
  5. Figure 6 has two typos (chemotheraly and prtoteins), also I would explain better /mention the Cyto C pathway

Author Response

Comments and Suggestions for Authors

The authors of the above manuscript have reviewed the recent researches on mitochondria and autophagy in the contest of resistance to chemotherapies. The review is very informative and well-constructed but I would like to suggest some revisions before acceptance for publication.

  1. Check English spelling and typos throughout the text:

Lane 35 I would suggest to start the sentence in a different way than “occurring through”;

Reply: We have revised the sentence as-

“It alters cancer cell signals that are needed for cellular growth or immune response stimulation through a wide range of molecular pathways.”

Lane 91 (;);

Reply: We have removed extra period.

Lane 67, 68 (different font in ref number, also the period is cryptic and needs to be adjusted or splitted in sentences);

Reply: We have adjusted font in ref number and the period.

Lane 102 (space and paragraph title); lane156 to 158 is a repetition of above sentence; Lane 200-202 is a repetition of above sentence and needs to be explained more the mechanism of action of cisplatin in the context of mt-DNA damage;

Reply: We have removed repeated sentence and adjusted font in ref number and the period.

We have added more about cisplatin in the context of mt-DNA damage as-

In cisplatin-treated cells, mtDNA damage is visible, and cells with depleted mtDNA derived from a cisplatin-sensitive ovarian cancer cell line developed resistance to cisplatin-induced cell death.

Paragraph 2.5 check references and substitute PMID numbers, also the first sentences are quite cryptic and I would suggest to adjust them and explain better the implication of PolG1 tumorigenesis and as a therapy;

Reply: We have removed PMID typos

We have revised first sentence as

‘’The only DNA polymerase known in human mitochondria, PolG, contains a large catalytic component, PolG1 (140 kDa), as well as two smaller identical accessory subunits, PolG2 (55 kDa), both of which are essential for embryonic development.’’

We have explained the implication of PolG1 tumorigenesis and as a therapy  as-

“The POLG1 has been shown to have altered genetic and epigenetic regulation in human malignancies, implying that POLG1 germline variations may play a role in tumorigenic characteristics [98]. For examples, the breast tumorigenesis is aided by mutations in mitochondrial DNA POLG1 [96]. It is also a crucial component in mtDNA replication and repair [99,100]. It has been reported that targeting mi-tochondrial DNA polymerase gamma with 2'3'-dideoxycytidinePolG could be a unique therapy method for acute myeloid leukemia [101,102]. For examples, in acute myeloid leukemia, the thymidine dideoxynucleoside analog alovudine inhibits mitochondrial DNA polymerase, hinders oxidative phosphorylation (OXPHOS), and pro-motes monocytic differentiation [101].’’

 Lane 229-230 role of RPS4X has to be explained better;

The role of The RPS4X has been explained as-

‘’X-linked ribosomal protein S4 (RPS4X), which is involved in cellular translation and proliferation in breast cancer cell lines, has been identified as a partner of the overexpressed multifunctional protein YB-1, and RPS4X depletion leads in consistent resistance to cisplatin (PMID: 21466612). It has been revealed that RPS4X deletion in ovarian cancer cell lines decreased their proliferative rate but more critically, increased their cisplatin re-sistance

Lane 304 I would use “drugs” rather than “medicines”. Also, I would check the entire paragraph in terms of English phrase construction and clarify in which types of tumors inhibition of OXPHOS can be used as a treatment and explain better how Warburg effect can be reversed and used for chemotherapy;

We have revised the sentence to better explain as-

‘The OXPHOS thought to be a process implicated in a variety of malignancies, and it is being in-vestigated as a possible target for cancer treatment. Blocking mitochondrial biogenesis, utilizing drugs that impact mitochondrial function, or using direct high-affinity inhibitors of respiratory chain complexes are all options for inhibiting OXPHOS. The OXPHOS inhibitors are used in various tumors, such as metformin for prostaste cancer, phenformin for melanoma, carboxyamidotriazole orotate for glioblastoma and atovaquone for acute promyelocytic leukemia [174,175]. However, due to inherent or acquired resistance of cancer cells to OXPHOS suppression, there are only a few cases in which OXPHOS inhibition would be sufficient to permanently suppress tumorigenesis. Therefore, the combination thera-pies could be devised to enhance the efficacy of OXPHOS inhibition in cancer treatment. BRAF inhibition, for example, can be coupled with phenformin to slow the progression of melanoma, and the cKIT inhibitor imatinib can be used with VLX600 to decrease the growth of gastrointes-tinal stromal tumors in mice.’’

‘’The glioblastoma growth and proliferation are inhibited when PDK1 and epidermal growth factor receptor (EGFR) are targeted together, reversing the Warburg effect through decreased HIF-1 expression, shifting the Warburg phenotype to OXPHOS, and inhibiting glioblastoma multiforme growth and proliferation.’’

Lane 315 adjusts the sentence;

Reply: We have adjusted the sentence as-

‘’The proportional contribution of glycolysis and OXPHOS to ATP production differs substantially between cell types; for example, the typical contribution of OXPHOS to ATP synthesis in normal cells is 80%, but it is less than that in cancer cells.’’

Lane 351 which anticancer drugs?

Reply: We have revised the sentence as-

 “Anticancer drugs that target thymidylate synthase and folate-dependent enzymes disturb metabolic pathways, which could lead to new cancer-fighting approaches.”

Lane 472 Figure 4 is repeated twice;

Reply: We have removed repeated Figure 4

Lane 502/570 reference needs to be adjusted PMID deleted;

Reply: We have deleted the PMID

Paragraph 5.1 I would suggest to avoid repetition and make the paragraph more concise and focused on the mitochondria fusion and fission. Also, I would describe more how mitochondria dynamics has been implicated in chemoresistance. In addition, several PMID references have to be corrected;

Reply: We have try to concise the paragraph by removing repeated sentence and focused on the mitochondria fusion and fission. And also corrected the PMID.

Lane 767-768 sentence is not complete, check PMID typos in the paragraph. Furthermore, I suggest to add more data on the mitochondrial potassium channels and their implication in cytoprotection;

Reply: We have completed the sentence 767-768 as-

‘’ATG12 conjugation regulates mitochondrial homeostasis and cell death, whereas altering the ATG12-ATG3 complex leads to increased mitochondrial mass, fragmentation of the mitochondrial network, and resistance to cell death mediated by mitochondrial pathways.’’

We have removed PMID typos

We have added more data on the mitochondrial potassium channels and their implication in   cytoprotection as-

  ‘’For example, in cardiac and neural tissue, activation of mitochondrial potassium chan-nels, such as ATP-regulated or calcium-activated large conductance potassium channels, may have cytoprotective effects (PMID: 32824877). It is critical to design a therapy that encourages cancer cells to undergo apoptosis in spite of such limitations, like Bax or Bak, which causes apoptosis resistance and limits the activity of chemotherapeutics. The Bax protein blocks the mitochondrial Kv1.3 potassium channel, making it a potential thera-peutic target (PMID: 31854954). For example, three inhibitors (Psora-4, PAP-1, and clo-fazimine) of mitochondrial Kv1.3 channels cause cancer cells to die in a Bax/Bak-independent manner (PMID: 22496117).’’

  1. Add appropriate references to the sentences referred to the following lanes: 42, 43,51,54,82,93,96,101,143,151,153,164,170,197,263,266,272,283,328,455,554,774,776,800

Reply: We have added relevant references to mentioned sentences.

  1. Figure 1: I would explain the figure legend and the mitochondrial components;

Reply: we have explained figure 1 legend and mitochondrial components as-

            “’The mitochondrial components are being targeted by chemotherapeutics that regulate metabolic activity, enabling for the selective killing of cancer cells for therapeutic benefit. To achieve therapeutic benefit the calcium channels or other ion complexes are more pominately targeted by chemotherapy, such as voltage-dependent anion channel (VDAC), mi-tochondrial calcium uniporter (MCU) complex, and mitochondrial permeability transition pore complex (mPTPC) etc. Chemotherapy also targeted various mitochondrial enzymes or proteins, including mitochondrial ribosome and ri-bosomal proteins (MRP), DNA polymerase subunit gamma (PolG) and ATP synthase etc. The mitochondrial DNA (mtDNA) and mitochondrial reactive oxygen species (mROS) considered to be a potential mitochondrial components for chemotherapy. ‘’

  1. Figure 3: It is worth adding mechanism of action of drugs (listed in table 1) on each component of the glycolysis pathway. I would suggest to make a parallel figure or to change Figure 3 to add a cancer cell in addition to a normal cell;

Reply: We have added the mechanism of action of drugs (listed in table 1) on each component of the glycolysis and OXPHOS in Figure 3.

  1. Figure 5: Figure legend has to be adjusted in terms of English using a more formal language;

Reply: We have adjusted some sentence using formal language as-

‘’The mitochondria increases their population by biogenesis process through utilizing fusion to fuse two neighboring mitochondria and fission to splits two mitochondria from one, whereas the mitophagy (selective autophagic machinery) selectively removes damaged mitochondria to maintain cellular homeostasis’’

  1. Figure 6 has two typos (chemotheraly and prtoteins), also I would explain better /mention the Cyto C pathway

Reply: We have corrected the typographical error. And explained the cyto C pathway as-

‘’Inhibition of mitochondrial Ca2+ signaling can block the release of mitochondrial cytochrome c (Cyto C), followed by suppression of interaction between Cyto C and apoptosome, subsequently inhibition of apoptosis pathway.’’

Reviewer 3 Report

The revised version of the manuscript titled “Chemotherapy Resistance: Crosstalk between Mitochondria and Autophagy” has addressed the previously raised aspects comprehensively.

The authors present an impressive abundance of relevant literature on the topic of the review and have devised numerous detailed and informative figures to illustrate the different chapters of their review. While significantly improved over the original version, the text still is oftentimes enumerating and gets lost in the details. The manuscript should undergo English editing assistance for linguistic clarity.

Author Response

The revised version of the manuscript titled “Chemotherapy Resistance: Crosstalk between Mitochondria and Autophagy” has addressed the previously raised aspects comprehensively.

The authors present an impressive abundance of relevant literature on the topic of the review and have devised numerous detailed and informative figures to illustrate the different chapters of their review. While significantly improved over the original version, the text still is oftentimes enumerating and gets lost in the details. The manuscript should undergo English editing assistance for linguistic clarity.

Reply: Thank you very much for your valuable comments and suggestions.  We have sent our manuscript to professional English editing system before submitting manuscript.

This manuscript is a resubmission of an earlier submission. The following is a list of the peer review reports and author responses from that submission.

Round 1

Reviewer 1 Report

This review covers a timely topic on the recent developments in the field of mitochondrial research, mitophagy, and their implications in chemotherapy resistance. The authors discuss the current knowledge about mitochondria proteins and how they could be regarded as pharmacological targets, considering novel mitochondria-targeted anti-cancer therapeutic approaches. Then the review discusses also the autophagic process, examining its role in cancer treatment. Finally, the authors discuss the connection between chemotherapy resistance, mitochondrial dysfunction, and autophagic machinery.

The review is written quite well and

However, I have some major concerns regarding the readability of the manuscript (see comments below):

- As it stands the review is hard to read and the topics are difficult to follow. The work reads more like a list of findings, rather than a critical review of the major findings put in context. The authors are encouraged to work on the readability of the work.

- Authors discuss the role of mitochondrial calcium only at the end of the review. They consider revising section 2 “Materials and Methods Potential Conventional Mitochondrial Targets for Chemotherapy” adding a discussion about how mitochondrial-resident proteins can regulate mitochondrial Ca2+ homeostasis affecting the sensitivity towards chemotherapeutic treatments.

- the review would benefit from a list of the numerous abbreviations used

- The figure legends should be more exhaustive, and authors should provide a list of abbreviations for the proteins shown in the figure.

-Authors should consider revising and expanding the section about autophagy. For example, lines 386-395 and lines 678-689 can be moved in this part.

- Lines 785-799 what do the authors mean?

Reviewer 2 Report

The manuscript titled “Chemotherapy Resistance: Crosstalk between Mitochondria and Autophagy” provided a summary of research on mitochondria and autophagy in chemoresistance. This should have been an interesting topic to review. However, the manuscript is not written and organized in a way that the readers can easily comprehend and grasp the key information. The overall background introduction is adequate, but the information is often too detailed and off-topic, making it difficult to follow. The logic and the clarity of writing require substantial improvements. The authors may require the assistance of English editing.  In below, the reviewer has provided specific comments and suggestions.

Comments

  1. Line 63: text “Material and Methods” was inserted.
  2. Line 65-78: The authors made a brief introduction on the mitochondria but it does not explain why these components make mitochondria is an excellent pharmacological target.
  3. How does the simultaneous treatment with pore inhibitor + pore opener sensitizer trigger cell death? We need specifics.
  4. Line 99-101, 103-105: The sentence “The ATP synthase…..breast cancer therapy” is duplicated.
  5. Line 109: It should be FOF1 ATP synthase, not F0F1.
  6. Line 161: The term “mitosome” commonly refers to a type of degenerated mitochondria in anaerobic eukaryotes. It should not be confused with mitochondrial ribosomes.
  7. Line 189: The sentence does not make sense. The purpose of chemotherapy is to eliminate cancerous cells, not just to “induce apoptosis via targeting Bcl-2 proteins”.
  8. Figure 1: There is an arrow point from “nucleus” to “mtDNA”. What does that mean? The author should include figure legends.
  9. Line 201: In section 3, the authors discussed several pharmacological approaches that inhibit glycolysis, a process that occurs in the cytosol. Thus, the title “Mitochondria-targeted anti-cancer therapeutic approaches” does not seem appropriate.
  10. Line 211: Figure 2A cannot be found. Also, there are two “Figure 2” in the manuscript.
  11. Line 253: “mitochondrial respiration chain” and “electron transport chain” are the same process.
  12. Line 256-259: The sentence “It has been demonstrated……necessarily diminished” is very confusing and should be rephrased.
  13. Line 165-291: The detail on how the electron transport chain works is irrelevant to the topic discussed. Thus it should be removed.
  14. Page 12: There is too much unnecessary detail on the regulatory mechanism of autophagy. The authors should focus on the pathways relevant to the topic discussed.
  15. Line 445: The meaning of “tumor cells need to use their organelles to maintain growth” is unclear.
  16. Line 453: what is the “GC cell”?
  17. Autophagy has context-dependent roles in cancer. Thus, the author should organize and summarize the research progress on this topic. The authors only provided limited evidence in the manuscript.
  18. Line 512: Why does apoptosis facilitate mtDNA segregation during mitosis and mitochondrial elimination?
  19. Line 533-535: Why OPA1 upregulation confers resistance to cytochrome C release? The authors need to be more specific.
  20. Line 539: The word “mitofission” is not a standard term for mitochondrial fission.
  21. Line 557: The compound should be “ABT-737”. The author should double-check the spelling.
  22. PINK1 and BNIP3 are known to have anti-tumor roles. This should be discussed.
  23. Line 640: How autophagy contributes to the efficacy of chemotherapy via apoptosis? The meaning is not clear.
  24. Line 645: How organelle and protein recycle can avoid DNA damage? It is very confusing.
  25. Lin 671: Unified autophagy-related gene names should be used. “Apg16” should be “atg16”.
  26. Line 670-690: The description of the autophagy mechanism does not seem fitting.
  27. Line 785-799: The authors should be very careful not to copy-paste the irrelevant text to the manuscript.
  28. The authors did not discuss the interaction between mitochondria and autophagy. Thus, the title “cross-talk” does not seem fitting.